

# Formalized classification of ephemeral wetland vegetation (Isoëto-Nanojuncetea class) in Poland (Central Europe)

Zygmunt Kącki[1,*], Andrzej Łysko[2], Zygmunt Dajdok[3], Piotr Kobierski[4], Rafał Krawczyk[5], Arkadiusz Nowak[6], Stanisław Rosadziński[7,8] and Agnieszka Anna Popiela[9,*]

[1] Botanical Garden, University of Wrocław, Wrocław, Poland
[2] Faculty of Computer Science and Information Technology, Western Pomeranian University of Technology in Szczecin, Szczecin, Poland
[3] Institute of Environmental Biology, University of Wrocław, Wrocław, Poland
[4] Zespół Szkół Technicznych im. Władysława Reymonta w Lubsku, Lubsko, Poland
[5] Institute of Biological Sciences, Maria Curie-Skłodowska University, Lublin, Poland
[6] Botanical Garden, Center for Biological Diversity Conservation, Polish Academy of Sciences, Warszawa, Poland
[7] Department of Plant Ecology and Environmental Protection, Faculty of Biology, Adam Mickiewicz University in Poznań, Poznań, Poland
[8] Current Affiliation: Poznań, Poland
[9] Institute of Biology, University of Szczecin, Szczecin, Poland
* These authors contributed equally to this work.

Corresponding author
Agnieszka Anna Popiela,
popiela@univ.szczecin.pl

## ABSTRACT

Formalized classification of the class Isoëto-Nanojuncetea has not been performed in Poland. We used 69,562 relevés stored in Polish Vegetation Database. Based on the literature and expert knowledge we selected 63 diagnostic species for the Isoëto-Nanojuncetea class. Unequivocal classification was applied in this work according to Cocktail method. A set of formal definitions was established using a combination of logical operators of total cover of species in case of high-rank syntaxa while sociological species groups and cover of particular species were used for logical formulas describing class, alliances and associations. An Expert System was prepared and applied to classify the whole data set of PVD and 1,340 relevés were organized at the class level. We stratifies the data and finally we used data set of 903 relevés to prepare synoptic tables, distribution maps and descriptions of the syntaxa. Twelve associations and two plant communities were identified. Vegetation of the Isoëto-Nanojuncetea class occur in Poland's central and southern part, with scattered stands in northern region. We described two new plant communities within Eleocharition and Radiolion alliance. The first formal classification of the Isoëto-Nanojuncetea class revealed a high diversity of ephemeral vegetation wetland found in Poland in the eastern boundary of their geographical distribution in Europe.

# INTRODUCTION

The classification of plant communities in Poland is not fully conducted, although studies on vegetation have a long tradition (*Matuszkiewicz, 2007*; *Kącki, Czarniecka & Swacha, 2013*).

There are syntaxonomic units, due to seasonal dynamics, unique habitat requirements and species-poor layout, that have not been defined. Vegetation of the class Isoëto-Nanojuncetea is one example. These annual, pioneer ephemeral dwarf-cyperaceous vegetation, occurs in Central Europe in mid-field hollows, at the bottoms of drying puddles, drained ponds, oxbows, lakes and rivers banks, in regularly flooded places, available for vegetation for a short period. Many species found on these locations are rare, although they usually have a wide distribution (*Deil, 2005*; *von Lampe, 1996*). The communities of Isoëto-Nanojuncetea class in Europe were first identified and reported at the beginning of the 20th century (*Koch, 1926*; *Libbert, 1932*; *Braun-Blanquet, 1931*, *1936*). Later on, an essential contribution to the knowledge of diversity and classification of Isoëto-Nanojuncetea in Europe was provided by *Moor (1936*, *1937)*, *Pietsch (1965*, *1969*, *1973)*, *Pietsch & Müller-Stoll (1968*, *1974)*, *Rivas Goday (1970)*, *Tüxen (1973)*, *Popiela (1997)*, *Brullo & Minissale (1998)*, *Täuber (2000)*, *Deil (2005*, *2020)* and *Šumberová & Hrivnák (2013)*.

The species diversity and communities's structure of ephemeral vegetation often depend on the length of time of ground exposure or agricultural intensity and the soil seed bank (*e.g. Poschlod et al., 2013*). At the initial development stage, an impoverishment in diagnostic species is often observed or the species are found in small patches among other annual vegetation, mostly of Bidentetea and Papaveretea classes. Additionally, some studies indicated that the same species are linked to different and distinctively defined plant associations. Hence, the classification of ephemeral wetland vegetation is confusing both on the level of associations and the whole class (*Popiela, 2005*; *Šumberová & Hrivnák, 2013*).

Plant communities of Isoëto-Nanojuncetea class in Poland reach out to the north-eastern distribution boundaries, and most of these diagnostic species are rare and vanishing (*Popiela, 2005*, *Kaźmierczakowa et al., 2016*). For the first time in Poland, they were described by *Kornaś (1960)*, *Wójcik (1968)* and *Zając & Zając (1988)*. The first comprehensive classification based on large dataset was proposed by *Popiela (1997)*. However, in recent decades, new regional studies on the syntaxonomy and ecology of Isoëto-Nanojuncetea vegetation have been published (e.g. *Spałek & Nowak, 2006*; *Krawczyk et al., 2016*). The new communities were identified and reported (*Fabiszewski & Cebrat, 2003*; *Dajdok, 2009*). All this data are stored in a national phytosociological database—the Polish Vegetation Database (*Kącki & Śliwiński, 2012*). Large vegetation-plot databases are currently available worldwide (*Dengler et al., 2011*; *Chytrý et al., 2016*; *Nowak et al., 2017*), making possible a comprehensive classification of vegetation at different spatial scales (*Dengler et al., 2013*; *De Cáceres & Wiser, 2012*). To achieve consistency across large regions, the Cocktail method is advised by *Bruelheide (1997*, *2000)* to classify different vegetation types in Europe (e.g. *Šilc & Čarni, 2007*; *Douda et al., 2016*; *Willner et al., 2019*; *Bruelheide et al., 2019*). The formalized approach uses definitions, by which a particular relevé is classified into vegetation units in a reproducible way. Still, the classification results are independent of the origin of the data set (*Bruelheide & Chytrý, 2000*). The first attempt in the classification of Isoëto-Nanojuncetea vegetation by the use of the Cocktail method was made by *Šumberová & Hrivnák (2013)* for Czech and Slovakia territory. The authors have described formal definitions at the association level. In this

study, we applied multilevel formalized classification (*Janišová & Dúbravková, 2010*), which was successfully introduced in classification of *Molinia* meadows in Poland (*Swacha, Kącki & Załuski, 2016*). This works aims to solve the inconsistencies in the Polish classification of ephemeral vegetation. We provide a formalized classification procedure and synthesis of Isoëto-Nanojuncetea class units for alliance and associations level. The classification considers the accepted and valid syntaxa described formerly in Europe as much as possible.

## MATERIALS AND METHODS

### Data set and classfication method

The 69,562 relevés stored in the Polish Vegetation Database (PVD) (*Kącki & Śliwiński, 2012*) were used for this study. Based on the literature and expert knowledge (*Oberdorfer, 1977*; *Popiela, 1997*; *Matuszkiewicz, 2007*; *Kącki, Czarniecka & Swacha, 2013*), we selected 63 diagnostic species for the Isoëto-Nanojuncetea class (Table 1). Then we defined groups of diagnostic species into class and alliances as reported from Central Europe (*Matuszkiewicz, 2007*; *Mucina et al., 2016*). Using groups of species, we created functional species groups based on the total cover of all species (#TC) separately for the class and alliances and functional species group with selection based on the highest cover of a single species (#SC) according to *Landucci et al. (2015)* approach. Sociological groups of species were simultaneously determined using the fidelity phi coefficient (*Chytrý et al., 2002*). We applied unequivocal classification in accordance with Cocktail method (*Bruelheide, 1997*). Formulas were created using the formal operators AND, OR, NOT available in Juice software (*Tichý, 2002*). The structure of the formulas for alliance level based on total cover groups #TC and cover of a single species #SC, while the association level was created by a combination of sociological species groups or a cover threshold of particular diagnostic species associated with the alliance`s total cover group, enabling us to keep the syntaxonomical hierarchy, which means that relevés classified to a particular association were also included into the appropriate alliance and the class (*Landucci et al., 2020*; *Kącki et al., 2020*). An Expert System was prepared and applied to classify the whole data set of PVD and 1,340 relevés were organized at the class level. Relevés without geographical coordinates and a surface area smaller than 1 m$^2$ or greater than 100 m$^2$ were excluded. We also removed cultivated plants from the analyses. Next, three relevés were randomly selected from each unit of grid square of 1.25 longitude × 0.75 latitude (ca. 1.5 × 1.4 km) from each vegetation units distinguished. Finally, we used data set of 903 relevés to prepare synthetic tables, distribution maps and descriptions of the syntaxa. For each syntaxon diagnosic, differential, constant and dominant species were determined and a combined synoptic table with percentage frequency and fidelity was created. Additionally, the statistical significance of the fidelity was calculated using Fisher's exact test (*Sokal & Rohlf, 1995*; *Chytrý et al., 2002*). We considered as a diagnostic only species which were included in the list of preselected 63 species for Isoëto-Nanojunceta class, while the others we indicated as differential species. Species were considered as diagnostic

**Table 1 List of sociological species groups used for formal definitions of syntaxa.**

| Species group | Species |
|---|---|
| # TC Isoëto-Nanojuncetea class | *Anagallis minima, Anthoceros punctatus s. l., Cardamine parviflora, Carex bohemica, Centaurium pulchellum, Cerastium dubium, Coleanthus subtilis, Crassula aquatica, Cyperus fuscus, Cyperus michelianus, Cyperus esculentus, Elatine alsinastrum, Elatine hexandra, Elatine hydropiper, Elatine triandra, Eleocharis acicularis, Eleocharis ovata, Fossombronia wondraczekii, Gnaphalium uliginosum, Gratiola neglecta, Gypsophila muralis, Hypericum humifusum, Illecebrum verticillatum, Isolepis setacea, Juncus bufonius, Juncus capitatus, Juncus ranarius, Juncus tenageia, Laphangium luteoalbum, Limosella aquatica, Lindernia dubia, Lindernia procumbens, Lythrum hyssopifolia, Lythrum portula, Mentha pulegium, Montia arvensis, Myosurus minimus, Phaeoceros laevis, Plantago major subsp. intermedia, Potentilla supina, Pulicaria vulgaris, Cyperus flavescens, Radiola linoides, Ranunculus sardous, Riccia beyrichiana, Riccia bifurca, Riccia canaliculata, Riccia cavernosa, Riccia ciliifera, Riccia crystallina, Riccia duplex, Riccia huebeneriana, Riccia warnstorfii, Riccia glauca, Riccia sorocarpa, Sagina apetala, Sagina nodosa, Schoenoplectus supinus, Spergularia echinosperma, Spergularia rubra, Veronica anagalloides, Veronica catenata, Veronica peregrina* |
| #TC Eleocharition soloniensis | *Carex bohemica, Coleanthus subtilis, Crassula aquatica, Cyperus fuscus, Cyperus michelianus, Elatine hexandra, Elatine hydropiper, Elatine triandra, Eleocharis ovata, Limosella aquatica, Lindernia dubia, Lindernia procumbens, Potentilla supina, Riccia canaliculata, Riccia cavernosa* |
| #TC Radiolion | *Anagallis minima, Anthoceros punctatus s.l., Fossombronia wondraczekii, Gypsophila muralis, Hypericum humifusum, Illecebrum verticillatum, Isolepis setacea, Juncus capitatus, Montia arvensis, Phaeoceros laevis, Radiola linoides, Ranunculus sardous, Spergularia rubra* |
| #TC Verbenion | *Elatine alsinastrum, Juncus ranarius, Lythrum hyssopifolia, Mentha pulegium, Pulicaria vulgaris, Cyperus flavescens, Schoenoplectus supinus* |
| ### *Cyperus fuscus* | *Cyperus fuscus, Potentilla supina, Limosella aquatica* |
| ### *Elatine alsinastrum* | *Alisma lanceolatum, Elatine alsinastrum, Schoenoplectus supinus* |
| ### *Elatine hexandra* | *Elatine hexandra, Elatine triandra, Elatine hydropiper* |
| ### *Eleocharis ovata* | *Carex bohemica, Eleocharis ovata, Lindernia procumbens* |
| ### *Myosurus minimus* | *Bryum ruderale, Myosurus minimus, Ranunculus sardous* |
| ### *Centunculus minimus* | *Anthoceros punctatus s. l., Anagallis minima, Juncus capitatus, Radiola linoides* |
| ### *Gypsophila muralis* | *Gypsophila muralis, Laphangium luteoalbum, Spergularia rubra* |
| ### *Isolepis setacea* | *Isolepis setacea, Juncus tenageia, Stellaria alsine* |

or differential for the alliances at phi ≥ 0.20 and for the association at phi ≥ 0.25 and Fisher's exact test at significance level of $p < 0.001$. Constant and dominant species were those with a frequency of ≥ 40% and with a cover value at least 25% in at least 5% of relevés of particular vegetation units, respectively. We used Euro-Med-Plant+ nomenclature for species names (access 11.2018). Taxonomically critical taxa were merged into aggregates.

## Statistics, map preparation, nomeclature

Ecological gradients of defined associations were evaluated by Detrended Correspondence Analysis (DCA) using Juice 7.1.28 and ordination methods available in R for Windows with VEGAN module (*Oksanen et al., 2011*). For interpretation of the main environmental gradients, average non-weighted Ellenberg indicator values for light, temperature, continentality, soil reaction, moisture and nutrients (*Ellenberg et al., 1992*) were used as supplementary variables. Ellenberg indicator values were calculated for weighted relevés by species cover (CWM). The ordination (DCA) was conducted with data transformation $b = (X_{i,j})^p$ ($p = 0.5$) and scaling by 26 segments. Distribution maps of all associations were prepared using PostgreSQL 11.0 database with PostGIS 2.6 extension. Visualisation

**Table 2 Number of relevés of identified plant communities of the Isoëto-Nanojuncetea class in Poland.**

| Cluster | Syntaxon | No. of relevés |
|---|---|---|
| A | Eleocharition soloniensis Philippi 1968 | 272 |
| 1 | Polygono-Eleocharitetum ovatae Eggler 1933 | 129 |
| 2 | Cypero fusci-Limoselletum aquaticae Oberd. ex *Korneck, 1960* | 127 |
| 3 | Cyperetum micheliani *Horvatić, 1931* | 4 |
| 4 | Community with *Coleanthus subtilis* | 12 |
| B | Verbenion supinae Slavnić 1951 | 46 |
| 5 | Veronico anagalloidis-Lythretum hyssopifoliae Wagner ex Holzner 1973 | 3 |
| 6 | Cyperetum flavescentis *Koch, 1926* | 8 |
| 7 | Pulicario vulgaris-Menthetum pulegii Slavnić 1951 | 19 |
| 8 | Eleocharito-Schoenoplectetum supini Soo & *Ubrizsy, 1948* nomina inversa prop. | 16 |
| C | Radiolion linoidis *Pietsch, 1973* | 223 |
| 9 | Stellario uliginosae-Isolepidetum setaceae *Libbert, 1932* | 20 |
| 10 | Centunculo minimi-Anthoceretum punctati Koch ex *Libbert, 1932* | 90 |
| 11 | Hyperico humifusi-Spergularietum rubrae *Wójcik, 1968* | 42 |
| 12 | Panico-Illecebretum verticillati Diemont et al., 1940. | 47 |
| 13 | Cerastio-Ranunculetum sardoi Oberdorfer ex Vicherek 1968 | 9 |
| 14 | Community with *Montia arvensis* | 15 |

of the result was prepared on QGIS3.16.3 and results showed on digital elevation map with accuracy $100 \times 100$ m by pixel.

# RESULTS

Three alliances with 14 plant communities were identified in Poland (Table 2). A total of 541 relevés were assigned to associations (60% of 903 relevés used) and 197 relevés were classified at the class level and 157 at the alliances level only (Table 2). We identified 12 plant associations and a community with *Coleanthus subtilis* placed into the Eleocharition soloniensis alliance, and a community with *Montia arvensis* included into the Radiolion linoidis alliance. According to DCA analysis based on the Ellenberg Indicator Values, the most influential factors determining the species composition are moisture and temperature (Fig. 1).

Communities of the Isoëto-Nanojuncetea class occur in the central and southern parts of Poland, so far they were not found in north-eastern Poland and are rarely found in northern parts (Fig. 2). Localities of the Eleocharition soloniensis stands are distributed mainly in the western and south-eastern part of Poland in big river valleys and fishpond regions (Fig. 3A). The Verbenion supinae communities are the rarest, sparsely distributed in Poland's middle part, mostly in the Lubelska Upland and the Wielkopolska Lowland (Fig. 3B). Localities of Radiolion linoidis alliance are distributed mainly in the central and the south-eastern part of Poland (Fig. 3C).

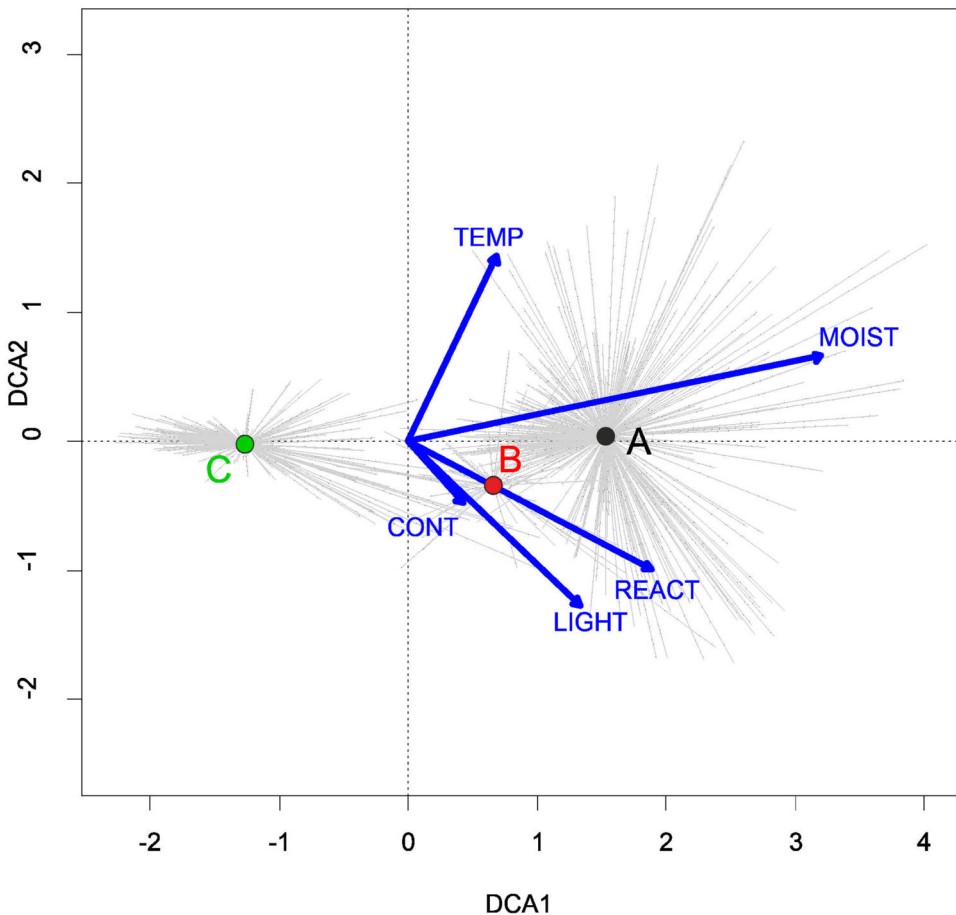

**Figure 1 DCA ordination diagram of samples of Isoëto-Nanojuncetea communities.** The first two ordination axes explain 34.8% and 36.6% total species-environment relations variability, and explain 3.0% and 5.2% of the entire species variability ($n$ = 541). The center of centroids marked by colored number. (A) Eleocharition ovatae. (B) Verbenion supinae. (C) Radiolion linoidis. The Ellenberg indicators—LIGHT, Light; TEMP, Temperature; CONT, Continentality; MOIST, Moisture; REACT, Soil Reaction; NUTR, Nutrient—were plotted as supplementary variables.

## List of syntaxa and species composition
### Eleocharition soloniensis Philippi 1968

**Diagnostic species:** *Eleocharis acicularis, Cyperus fuscus, Limosella aquatica, Eleocharis ovata, Carex bohemica, Elatine hexandra, Elatine hydropiper, Riccia cavernosa, Elatine triandra*

**Differential species:** *Callitriche palustris s. l., Alisma plantago-aquatica, Rorippa palustris, Leersia oryzoides, Juncus articulatus, Alopecurus geniculatus, Ranunculus sceleratus, Juncus bulbosus, Physcomitrium eurystomum*

**Constant species:** *Cyperus fuscus, Limosella aquatica, Gnaphalium uliginosum, Eleocharis acicularis, Rorippa palustris*

**Dominant species:** *Cyperus fuscus, Eleocharis acicularis, Eleocharis ovata, Limosella aquatica, Elatine hexandra*

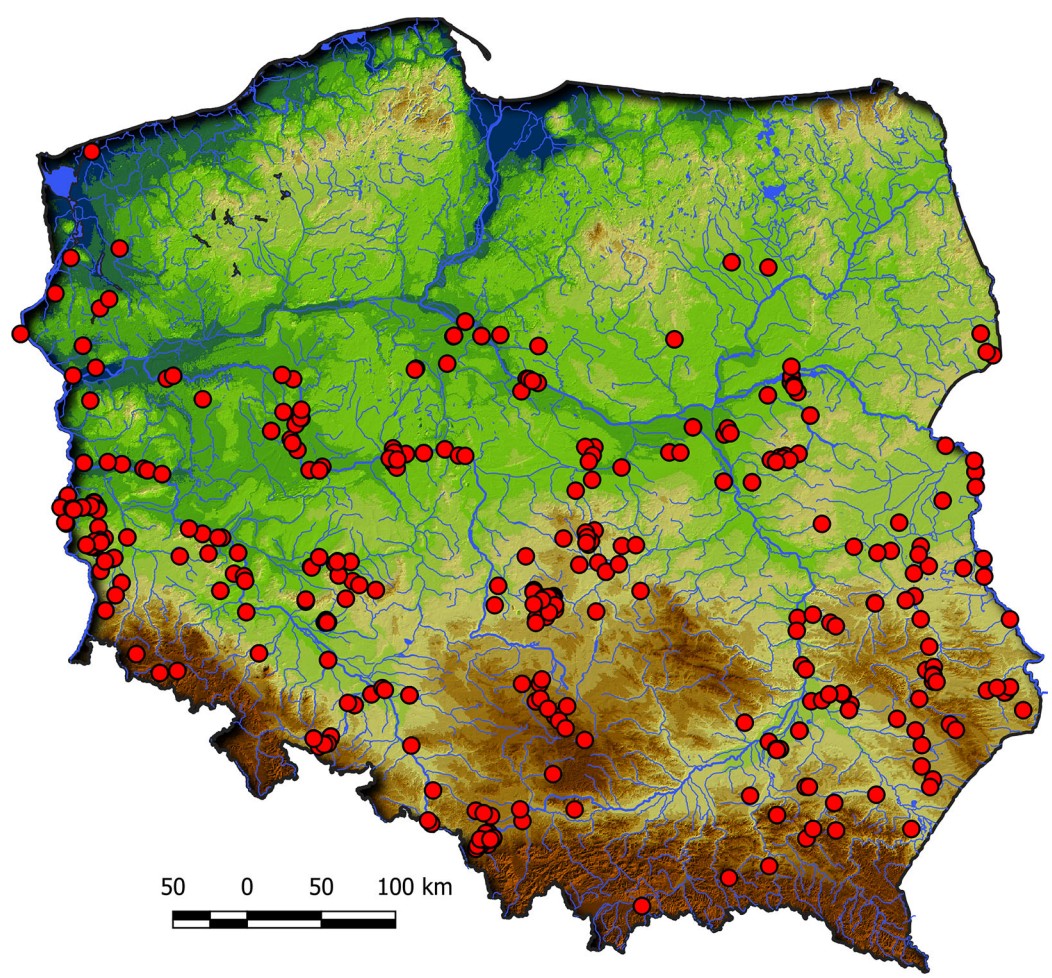

**Figure 2 Distribution of plant communities of the Isoëto-Nanojuncetea class in Poland.**

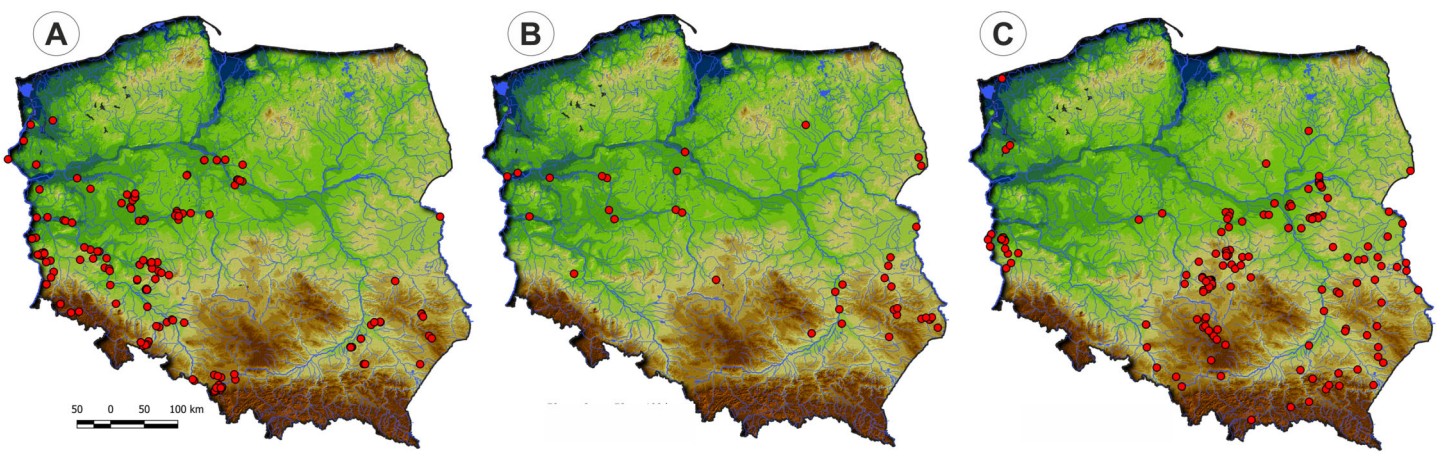

**Figure 3 The ranges of alliances of Isoëto-Nanojuncetea class in Poland.** (A) Eleocharition soloniensis. (B) Verbenion supinae. (C) Radiolion linoidis.

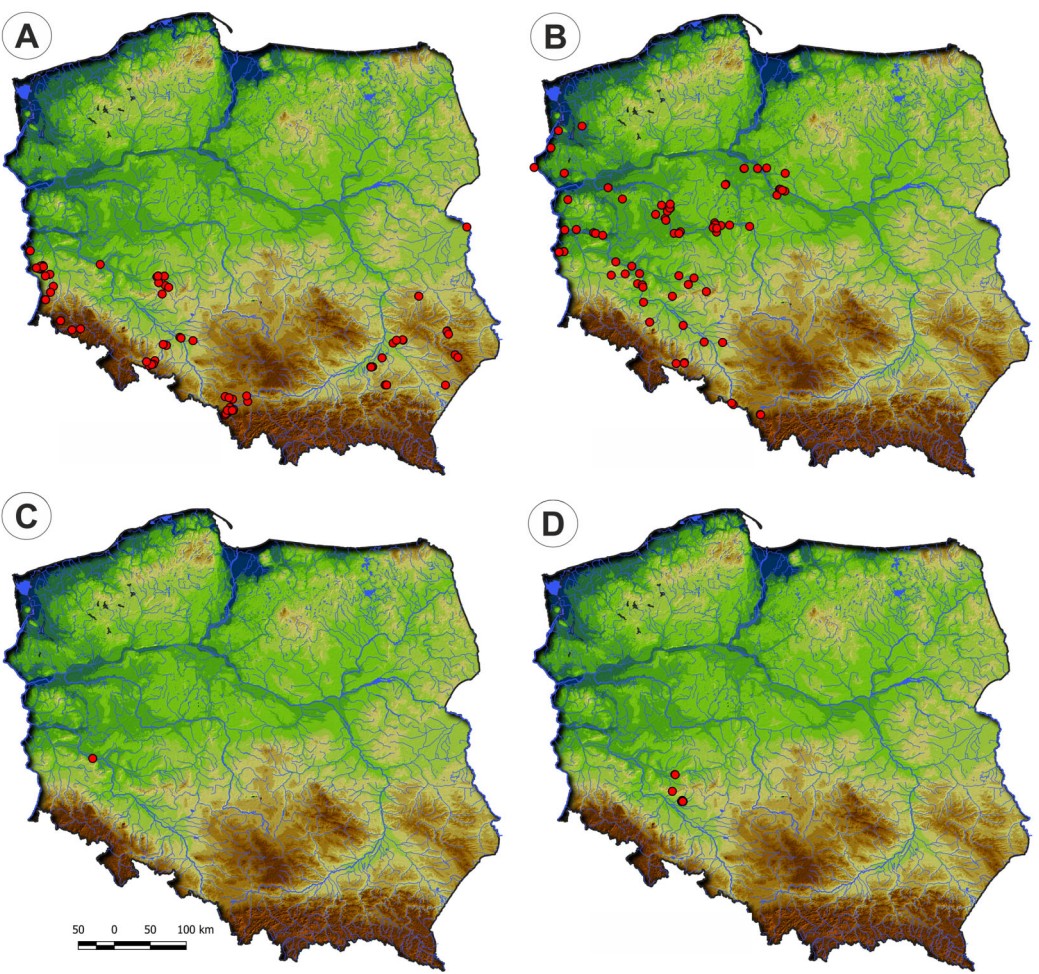

**Figure 4 Distribution of plant communities of the Eleocharition soloniensis alliance in Poland.**
(A) Polygono-Eleocharitetum ovatae. (B) Cypero-Limoselletum aquaticae. (C) Cyperetum micheliani.
(D) Community with *Coleanthus subtilis*.

In Poland, the Eleocharition soloniensis alliance comprises four plant communities (Appendix 2 and 3):

1. Polygono-Eleocharitetum ovatae Eggler 1933
2. Cypero fusci-Limoselletum aquaticae Oberd. ex *Korneck, 1960*
3. Cyperetum micheliani *Horvatić, 1931*
4. Community with *Coleanthus subtilis*

They develop on moderately fertile habitats, mainly on the drying-out bottom of fishponds, the margins of mid-field ponds, and periodically exposed banks of river channels, mostly in Poland's western part. (Fig. 4). According to DCA analysis, based on the Ellenberg Indicator Values, moisture, nutrients, and temperature differ the vegetation of the Eleocharition alliance (Fig. 5).

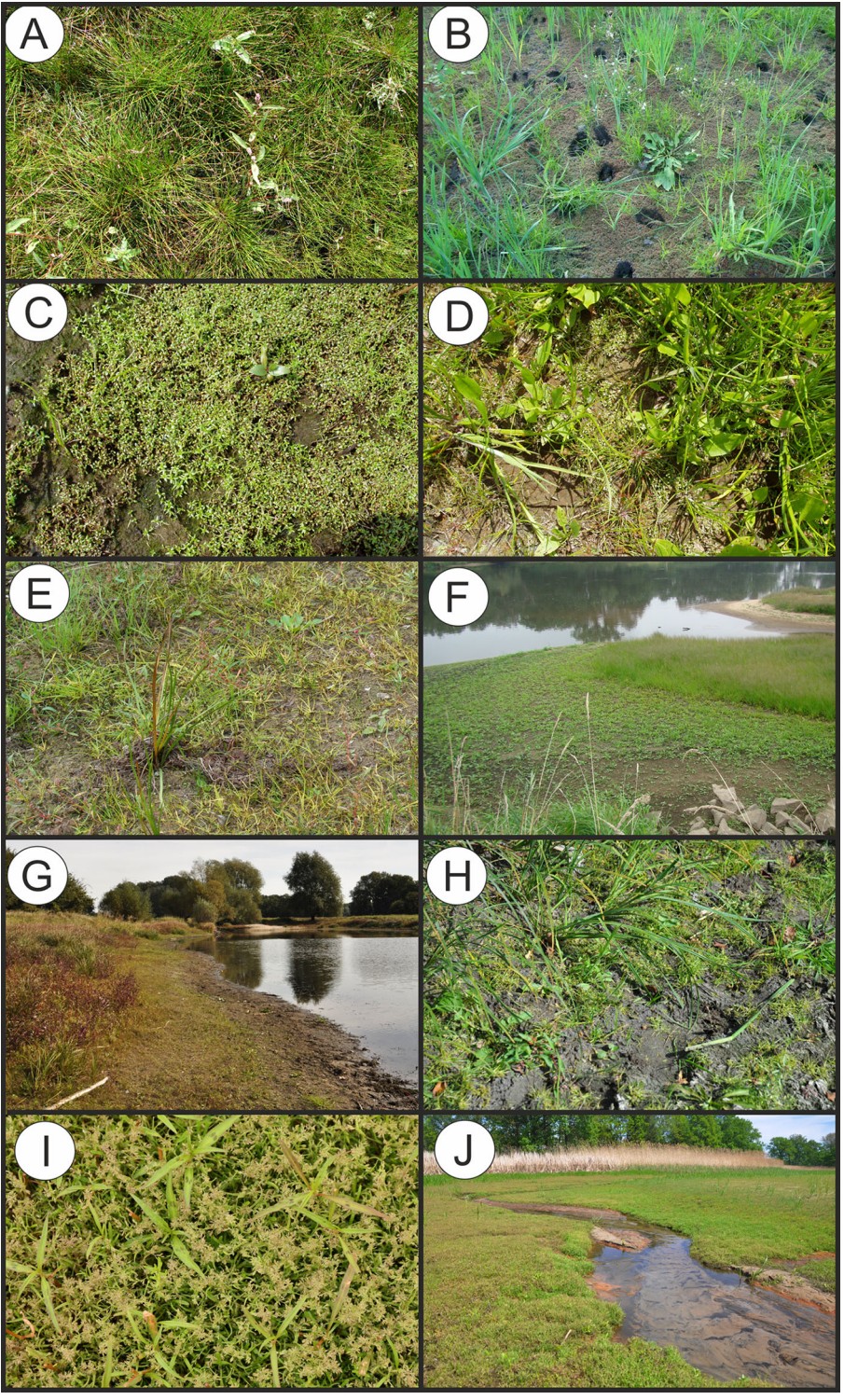

**Figure 5 Vegetation of the Eleocharition soloniensis alliance.** (A–D) Phytocoenoses of the Poly-gono-Eleocharetum ovatae in different stages of development (Photos by R. Krawczyk, S. Rosadziński, A. Popiela). (E and F) Phytocoenoses of Cypero-Limoselletum (Photos by S. Rosadziński). (G and H) Phytocoenoses of the Cyperetum micheliani (Photos by Z. Kącki). (I and J) Community with *Coleanthus subtilis* (Photo by Z. Dajdok).

## Polygono-Eleocharitetum ovatae Eggler 1933

**Diagnostic species**: *Eleocharis ovata, Carex bohemica, Elatine hexandra, Eleocharis acicularis, Elatine triandra, Lindernia procumbens*
**Differential species:** *Alisma plantago-aquatica, Juncus bulbosus, Physcomitrium eurystomum, Oenanthe aquatica, Scirpus radicans*
**Constant species**: *Eleocharis ovata, Eleocharis acicularis, Alisma plantago-aquatica,*
**Dominant species:** *Eleocharis ovata, Elatine hexandra, Eleocharis acicularis, Carex bohemica, Juncus bulbosus*

**Distribution:** Polygono-Eleocharetum ovatae occurs on scattered sites in the south-western and south-eastern parts of Poland. It does not occur in the north of the country due to scarcity of typical habitat—the fishpond complexes (Fig. 4).
**Physiognomical layout:** The individual patches show a single-layer structure and form a low turf-like vegetation, with varying coverage. The association often develops patchy structure, with typical plots composed of diagnostic species dispersed between stands of Phragmito-Magnocaricetea and Bidentetea vegetation (Fig. 5).
**Habitat requirements:** The Polygono-Eleocharitetum ovatae is associated with ponds habitats (old facilities), with regular, extensive fishery management for a long time. It does not appear on natural water bodies across Poland. It is mostly found on a fine-grained, wet, moist or drying sandy-loamy or silty substrate, rich in mineral compounds. Within the Eleocharition alliance this association is usually the most waterlogged one habitat (Fig. 6).

## Cypero fusci-Limoselletum aquaticae Oberd. ex *Korneck, 1960*

**Diagnostic species:** *Cyperus fuscus, Limosella aquatica, Eleocharis acicularis*
**Differential species:** *Salix purpurea, Plagiochila asplenioides*
**Constant species:** *Cyperus fuscus, Limosella aquatica, Gnaphalium uliginosum, Plantago major subsp. intermedia, Juncus bufonius, Rorippa palustris, Rumex maritimus*
**Dominant species:** *Cyperus fuscus, Limosella aquatica, Plantago major subsp. intermedia, Eleocharis acicularis*

**Distribution:** The Cypero fusci-Limoselletum aquaticae occurs mostly in the western part of Poland on natural sites along river valleys of Odra, Warta and lower part of Vistula while rarely is found on fishpond's complexes (Fig. 4).
**Physiognomical layout:** The association develops compact turf, usually at late summer. It is distinguished by the abundant occurrence of *Cyperus fuscus* (Fig. 5).
**Habitat requirements:** In Poland, patches of the Cypero fusci-Limoselletum aquaticae develop periodically on flooded habitats, located on gently sloping banks, coastal mudslides, margins of oxbow lakes and small water-filled hollows on the flooded terraces. They are also found on the freshly exposed bottom of water reservoirs, e.g. lakes, small mid-field depressions, at the bottoms of drained fishponds and dam reservoirs. This association has the highest requirements regarding nutrients and have less demand for moisture within the whole alliance' communities (Fig. 6).
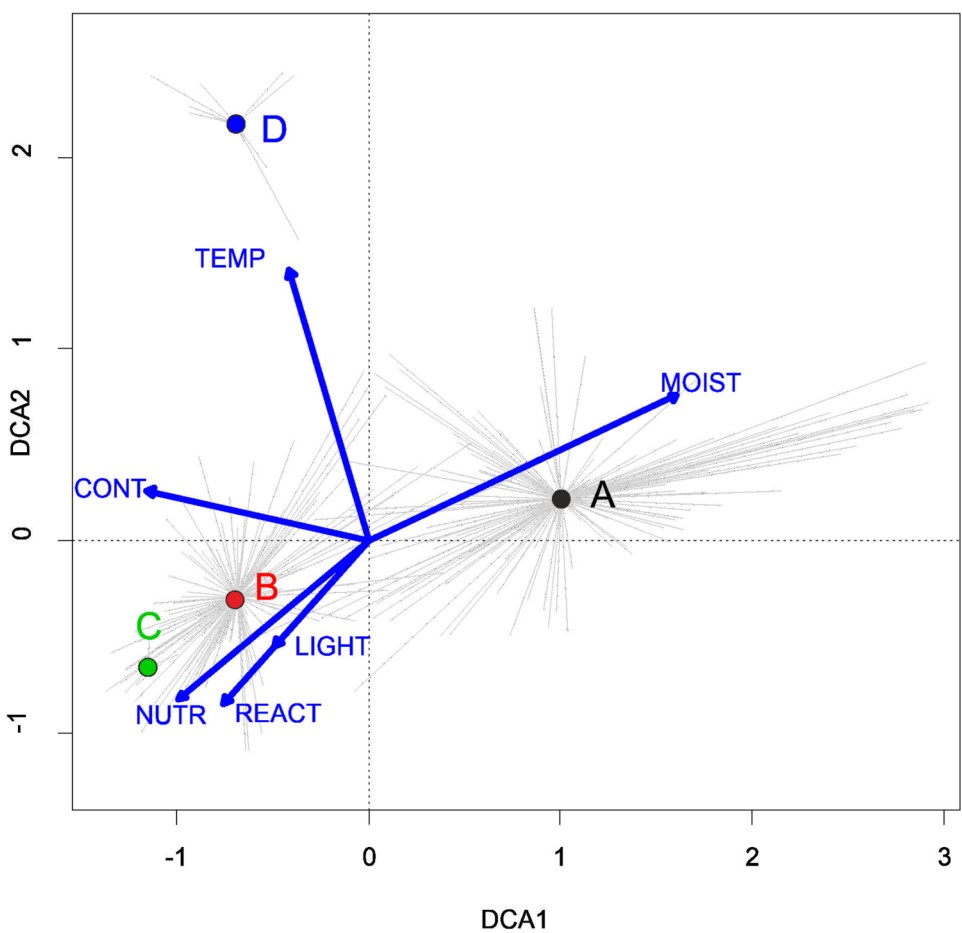

**Figure 6** **DCA ordination diagram of samples of Eleocharition soloniensis vegetation.** The first two ordination axes explain 23.9% and 27.8% of the total species-environment relations variability, and additional explain 3.8% and 6.7% of the total species variability ($n = 272$). Colored dots mark the center of centroids. (A) Polygono-Eleocharitetum ovatae. (B) Cypero-Limoselletum aquaticae. (C) Cyperetum micheliani. (D) community with *Coleanthus subtilis*. The Ellenberg indicators: LIGHT, Light; TEMP, Temperature; CONT, Continentality; MOIST, Moisture; REACT, Soil Reaction; NUTR, Nutrient were plotted as a supplementary variables.

### Cyperetum micheliani *Horvatić, 1931*

**Diagnostic species:** *Cyperus michelianus*

**Constant species:** *Rumex maritimus, Plantago major subsp. intermedia, Gnaphalium uliginosum, Cyperus michelianus, Cyperus fuscus, Rorippa palustris, Oxybasis rubra, Cirsium arvense, Salix fragilis, Rorippa amphibia, Lythrum salicaria, Limosella aquatica, Erigeron canadensis, Bidens frondosus, Atriplex prostrata, Agrostis stolonifera*

**Dominant species:** *Cyperus michelianus, Cyperus fuscus*

**Distribution:** The Cyperetum micheliani was found on single locality in south-western part of Poland, in the Odra river valley near Głogów (Fig. 4).

**Physiognomical layout:** The association develops compact turf and and is distinguished by the dominance of *Cyperus michelianus* (Fig. 5).

**Habitat requirements:** In Poland, patches of the Cyperetum micheliani develop on margins of oxbow lakes and on the flooded terraces of the riverbanks. The association has highest requirements regarding nutrients and has less demand for moisture within the Eleocharition alliance (Fig. 6).

## Community with *Coleanthus sublitis*

**Diagnostic species:** *Coleanthus subtilis, Veronica peregrina, Riccia cavernosa, Myosurus minimus, Limosella aquatica*

**Differential species:** *Ranunculus trichophyllus, Callitriche palustris s. l., Persicaria lapathifolia s. l.*

**Constant species:** *Coleanthus subtilis, Veronica peregrina, Myosurus minimus, Limosella aquatica, Persicaria lapathifolia s. l., Rorippa palustris, Callitriche palustris s. l., Persicaria hydropiper, Juncus bufonius, Riccia cavernosa, Ranunculus trichophyllus*

**Dominant species:** *Coleanthus subtilis, Veronica peregrina, Myosurus minimus, Riccia cavernosa, Persicaria lapathifolia s. l., Limosella aquatica, Callitriche palustris s. l., Ranunculus trichophyllus*

**Distribution:** So far, the community was found only in the south-western part of Poland, in fishpond complexes in Lower Silesia (Fig. 4).

**Physiognomical layout:** The community is well distinguished by the dominance of short grasses mostly the diagnostic species *Coleanthus subtilis* (Fig. 5).

**Habitat requirements:** Preferably, plots of is this community develop in ponds used in annual or biennial cycle (stocking ponds). So far, this plant communities developing in autumn or during seasonal summer water deficit have been recorded much less frequently.

## Verbenion supinae Slavnić 1951

**Diagnostic species:** *Pulicaria vulgaris, Cyperus flavescens, Elatine alsinastrum, Schoenoplectus supinus, Lythrum hyssopifolia*

**Differential species:** *Agrostis stolonifera, Alisma lanceolatum, Atriplex prostrata, Argentina anserina, Xanthium orientale subsp. italicum, Inula britannica, Cynosurus cristatus, Salix triandra, Poa pratensis*

**Constant species:** *Agrostis stolonifera, Plantago major subsp. intermedia*

**Dominant species:** *Pulicaria vulgaris, Elatine alsinastrum, Schoenoplectus supinus, Juncus bufonius, Cyperus flavescens, Plantago major subsp. intermedia, Alisma lanceolatum*

According to the formal classification of our data set, the Verbenion supinae alliance includes four associations (Appendix 2 and 3):

1. Veronico anagalloidis-Lythretum hyssopifoliae Wagner ex Holzner 1973
2. Cyperetum flavescentis *Koch, 1926*
3. Pulicario vulgaris-Menthetum pulegii Slavnić 1951
4. Eleocharito-Schoenoplectetum supini Soo & *Ubrizsy, 1948* (orig. *Schoenoplectus supinus-Heleocharis acicularis* assz.)

In Poland, vegetation of this alliance occurs rarely, mainly in the big river valleys and in the Lublin Upland (Fig. 3). According to DCA analysis based on the Ellenberg Indicator

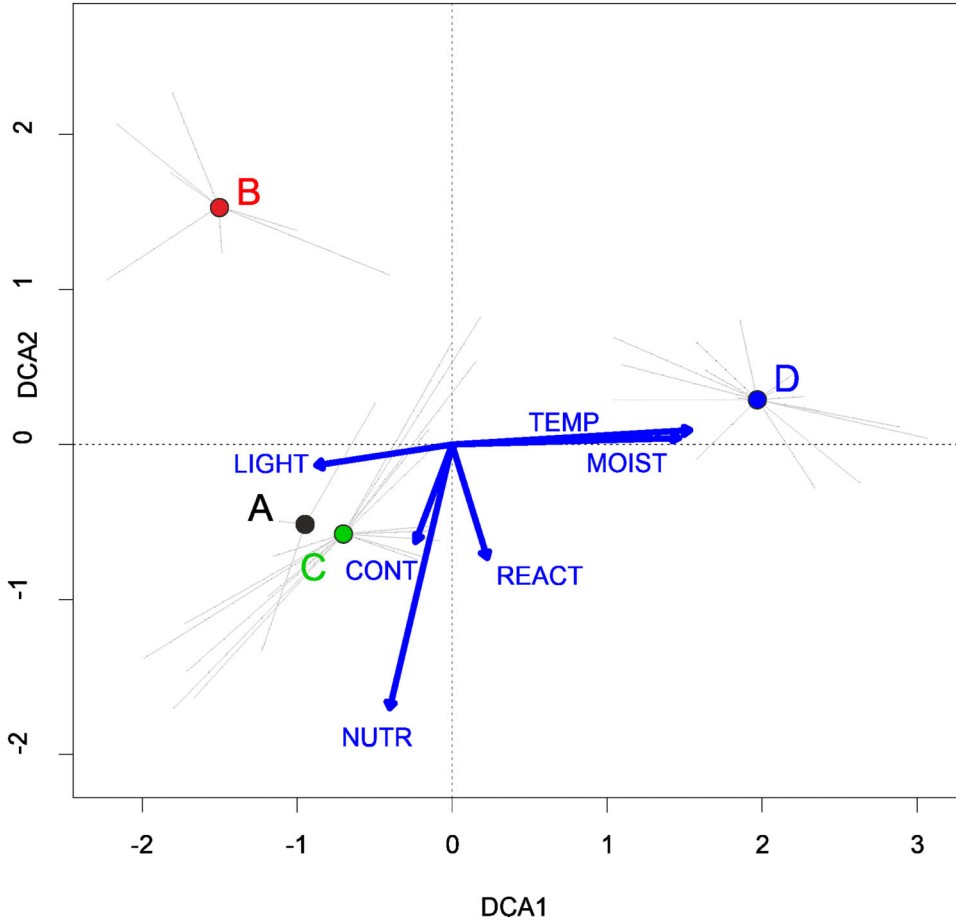

**Figure 7 DCA ordination diagram of samples of Verbenion supinae plant communities.** The first two ordination axes explain 21.2% and 32.7% of the total species-environment relations variability, and explain 8.0% and 14.4% of the total species variability ($n = 46$). (A) Veronico anagalloidis-Lythretum hyssopifoliae. (B) Cyperetum flavescentis. (C) Pulicario vulgaris-Menthetum pulegii. (D) Eleocharito-Schoenoplectetum supini. The Ellenberg indicators:- LIGHT, Light; TEMP, Temperature; CONT, Continentality; MOIST, Moisture; REACT, Soil Reaction; NUTR, Nutrient - were plotted as a supplementary variables.                                              

Values, Poland's Verbenion supinae alliance has high demand of the temperature and nutrient content (Fig. 7).

## Veronico anagalloidis-Lythretum hyssopifoliae Wagner ex Holzner 1973

**Diagnostic species:** *Juncus ranarius*
**Constant species:** *Juncus ranarius, Oxybasis rubra, Oxybasis glauca, Juncus compressus, Atriplex prostrata, Argentina anserina, Agrostis stolonifera*
**Dominant species:** *Juncus ranarius, Carex secalina, Bolboschoenus maritimus agg.*

**Distribution:** There are only three known sites of the Veronico anagalloidis-Lythretum hyssopifoliae in Poland located in the lowland belt of Wielkopolska, Kujawy and Mazury regions (Fig. 8).

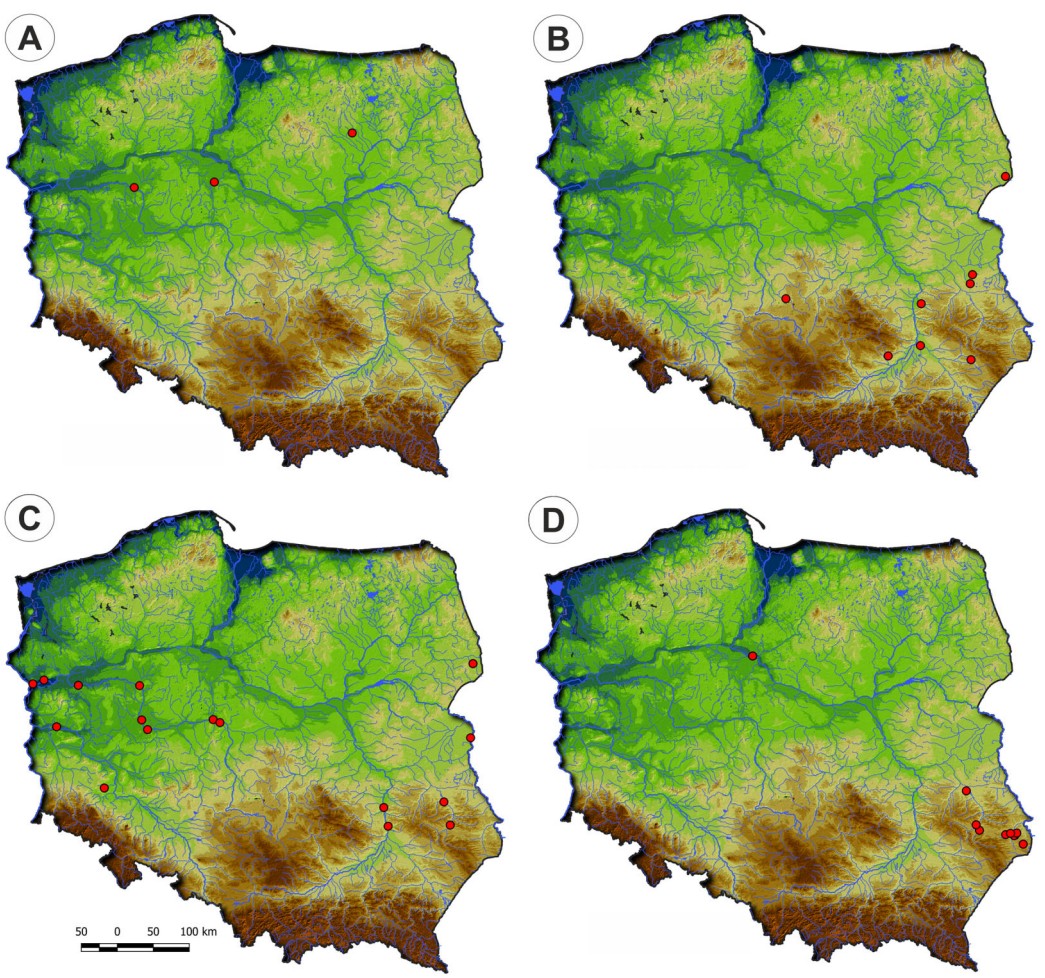

**Figure 8 Distribution of plant communities of the Verbenion supinae alliance in Poland.** (A) Veronico anagalloidis-Lythretum hyssopifoliae. (B) Cyperetum flavescentis. (C) Pulicario vulgaris-Menthetum pulegii. (D) Eleocharito-Schoenoplectetum supini.

**Physiognomical layout:** The association is well defined by *Juncus ranarius* and the presence of halophilous species.

**Habitat requirements:** The association occurs on saline, moist silts, and wet soils in nutrient rich habitats.

## Cyperetum flavescentis *Koch, 1926*

**Diagnostic species:** *Cyperus flavescens, Sagina nodosa*
**Differential species:** *Calliergonella cuspidata, Triglochin palustris, Ranunculus flammula*
**Constant species:** *Cyperus flavescens, Sagina nodosa, Ranunculus flammula, Juncus articulatus, Calliergonella cuspidata, Agrostis stolonifera*
**Dominant species**: *Cyperus flavescens*

**Distribution:** The Cyperetum flavescentis is distributed in the south-eastern part of Poland (Fig. 8).

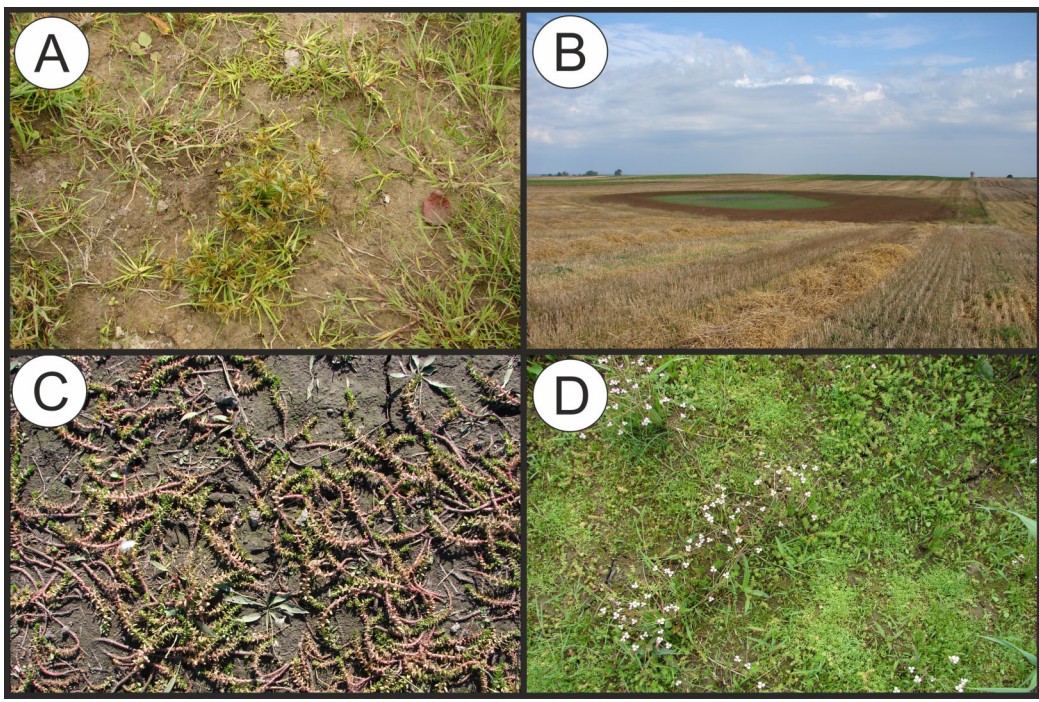

**Figure 9 Vegetation of the Verbenion supinae alliance.** (A) Phytocoenose of the Cyperetum flavescentis association. (B) a temporary pool in arable field depressions with patches of the Eleocharito-Schoenoplectetum supini (brown belt). (C and D) Phytocoenoses of the Eleocharito-Schoenoplectetum supini in different stages of development (Photo by R. Krawczyk).

**Physiognomical layout:** It forms very low and sparsy turfs on a sandy or sandy-loam substrates with weak acidity. The plant communities are usually clearly dominated by *Cyperus flavescens* (Fig. 9).

**Habitat requirements:** The communities develop periodically on flooded and grazed lakesides as well as peaty hollows. According to DCA analysis Cyperetum flavescentis requires lower humidity and nutrients content than other associations of the Verbenion alliance (Fig. 7).

## Pulicario vulgaris-Menthetum pulegii Slavnić 1951

**Diagnostic species**: *Pulicaria vulgaris*

**Differential species:** *Xanthium orientale subsp. italicum, Inula britannica, Lysimachia nummularia, Matricaria discoidea, Poa pratensis, Cynosurus cristatus, Agrostis stolonifera*

**Constant species**: *Pulicaria vulgaris, Agrostis stolonifera, Plantago major subsp. intermedia, Persicaria hydropiper, Juncus bufonius, Gnaphalium uliginosum, Persicaria lapathifolia s. l., Rumex maritimus, Argentina anserina, Bidens tripartita*

**Dominant species**: *Pulicaria vulgaris, Juncus bufonius, Plantago major subsp. intermedia, Eleocharis acicularis, Mentha pulegium, Lysimachia nummularia*

**Distribution:** In Poland, the association occurs on scattered sites, mainly in the valleys of Odra and Warta rivers (Fig. 8).

**Physiognomical layout:** It is a community found mostly in the flood zone of big rivers. In addition to the dominating diagnostic species, an essential contribution to this vegetation are species from the Bidentetea class.

**Habitat requirements:** The association reveals relatively high nutrients demand (Fig. 7).

## Eleocharito-Schoenoplectetum supini Soo & *Ubrizsy, 1948* nomina inversa prop. (orig. *Schoenoplectus supinus - Heleocharis acicularis assz.*)

**Diagnostic species**: *Elatine alsinastrum, Schoenoplectus supinus, Lythrum hyssopifolia*
**Differential species:** *Alisma lanceolatum, Persicaria amphibia*
**Constant species**: *Alisma lanceolatum, Elatine alsinastrum, Schoenoplectus supinus, Limosella aquatica*
**Dominant species**: *Elatine alsinastrum, Schoenoplectus supinus, Alisma lanceolatum, Plantago major subsp. intermedia, Lythrum portula, Juncus bufonius*

**Distribution:** Eleocharito-Schoenoplectetum supini is scattered in Poland. The majority of records come from the Lublin region's uplands, but it was also sporadically found in northwestern Poland (Fig. 8).

**Physiognomical layout:** In the initial phase, it is a distinguished pioneer vegetation type and was encountered in both the terrestrial and aquatic habitats. Its optimal phase is characterized by a high cover and relatively height (up to half a meter) (Fig. 9).

**Habitat requirements:** The association develops most often in temporary pools in arable field depression (some of them occur episodically during periods with abundant precipitation). The association is the most demanding considering moisture and temperature within the Verbenion supinae alliance (Fig. 7).

## Radiolion linoidis *Pietsch, 1973*

**Diagnostic species:** *Hypericum humifusum, Radiola linoides, Gypsophila muralis, Anagallis minima, Anthoceros punctatus s. l., Spergularia rubra, Juncus capitatus, Illecebrum verticillatum, Phaeoceros laevis, Gnaphalium uliginosum, Riccia glauca, Isolepis setacea, Riccia sorocarpa, Fossombronia wondraczekii, Montia arvensis*

**Differential species:** *Spergula arvensis, Rumex acetosella, Scleranthus annuus, Equisetum arvense, Sagina procumbens, Setaria pumila, Elytrigia repens, Viola arvensis, Fallopia convolvulus, Polygonum aviculare s. l., Mentha arvensis, Stellaria media s. l., Veronica arvensis, Juncus bufonius, Achillea millefolium, Veronica serpyllifolia, Erigeron canadensis, Anthemis arvensis, Raphanus raphanistrum, Anagallis arvensis, Myosotis arvensis, Cyanus segetum, Cerastium fontanum subsp. vulgare, Cirsium arvense, Apera spica-venti, Teesdalia nudicaulis, Stachys palustris, Vicia sativa s. l., Oxalis fontana, Arnoseris minima, Vicia hirsuta, Convolvulus arvensis, Galeopsis tetrahit s. l., Capsella bursa-pastoris, Holcus mollis, Ceratodon purpureus, Erodium cicutarium, Digitaria ischaemum, Agrostis capillaris, Bryum argenteum, Setaria viridis, Vicia tetrasperma, Veronica persica, Ranunculus repens, Pohlia nutans, Sonchus arvensis*

**Constant species:** *Juncus bufonius, Gnaphalium uliginosum, Rumex acetosella, Polygonum aviculare s. l., Plantago major subsp. intermedia, Sagina procumbens, Hypericum humifusum, Radiola linoides, Gypsophila muralis, Spergula arvensis, Anagallis minima, Spergularia rubra, Scleranthus annuus, Equisetum arvense*

**Dominant species:** *Juncus bufonius, Isolepis setacea*

There are six associations of the Radiolion linoidis in Poland distinguished by a formal classification (Appendix 2 and 3):

1. Stellario uliginosae-Isolepidetum setaceae *Libbert, 1932*
2. Centunculo minimi-Anthoceretum punctati Koch ex *Libbert, 1932*
3. Hyperico humifusi-Spergularietum rubrae *Wójcik, 1968*
4. Panico-Illecebretum verticillati Diemont et al., 1940
5. Cerastio dubii-Ranunculetum sardoi Oberdorfer ex Vicherek
6. Community with *Montia arvensis*

According to DCA analysis based on the Ellenberg Indicator Values, the Radiolion linoidis alliance communities are differentiated by habitat requirements in terms of moisture, light, nutrients, and soil reaction.

## Stellario uliginosae-Isolepidetum setaceae *Libbert, 1932*

**Diagnostic species**: *Isolepis setacea*

**Differential species:** *Stellaria alsine, Juncus articulatus, Leontodon saxatilis, Calamagrostis epigejos, Juncus effusus, Carex flava agg.*

**Constant species**: *Isolepis setacea, Juncus articulatus, Juncus bufonius, Gnaphalium uliginosum, Stellaria alsine, Sagina procumbens, Plantago major subsp. intermedia*

**Dominant species:** *Isolepis setacea, Juncus bufonius*

**Distribution:** Stellario uliginosae-Isolepidetum setaceae rarely occurs in Poland, mostly in the river valleys in its western part (Fig. 10).

**Physiognomical layout:** It occurs in the form of low turf with varying degrees of vegetation cover. The communities have a considerable number of diagnostic species of the Molinio-Arrhenatheretea and the Scheuchzerio-Caricetea vegetation (Fig. 11).

**Habitat requirements:** This plant community was recorded on extensively used forest roads, as well as mid-field depressions and pond margins. According to DCA results, Stellario uliginosae-Isolepidetum setaceae has higher demand of moisture and lower soils reaction requirements than other communities of the Radiolion alliance (Fig. 12).

## Centunculo minimi-Anthoceretum punctati Koch ex *Libbert, 1932*

**Diagnostic species**: *Anagallis minima, Anthoceros punctatus s. l., Radiola linoides, Juncus capitatus, Phaeoceros laevis,*

**Differential species:** *Rumex acetosella*

**Constant species**: *Anagallis minima, Juncus bufonius, Radiola linoides, Gnaphalium uliginosum, Anthoceros punctatus s. l., Polygonum aviculare s. l., Sagina procumbens,*

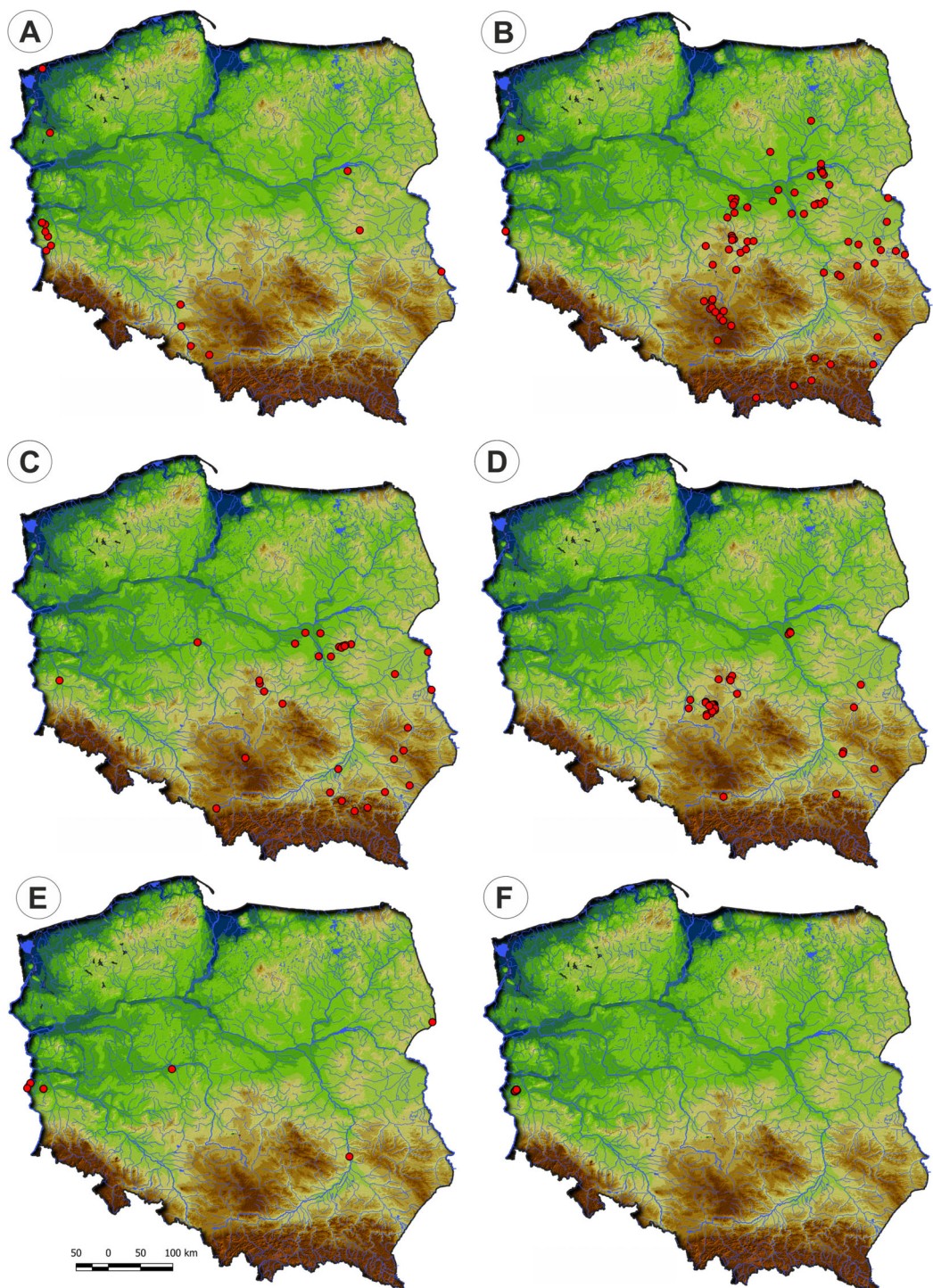

**Figure 10 Distribution of the plant communities of the Radiolion linoidis alliance in Poland.**
(A) Stellario uliginosae-Isolepidetum setaceae. (B) Centunculo minimi-Anthoceretum punctati.
(C) Hyperico humifusi-Spergularietum rubrae. (D) Panico-Illecebretum verticillati. (E) Cerastio dubii-Ranunculetum sardoi. (F) Community with *Montia arvensis*.

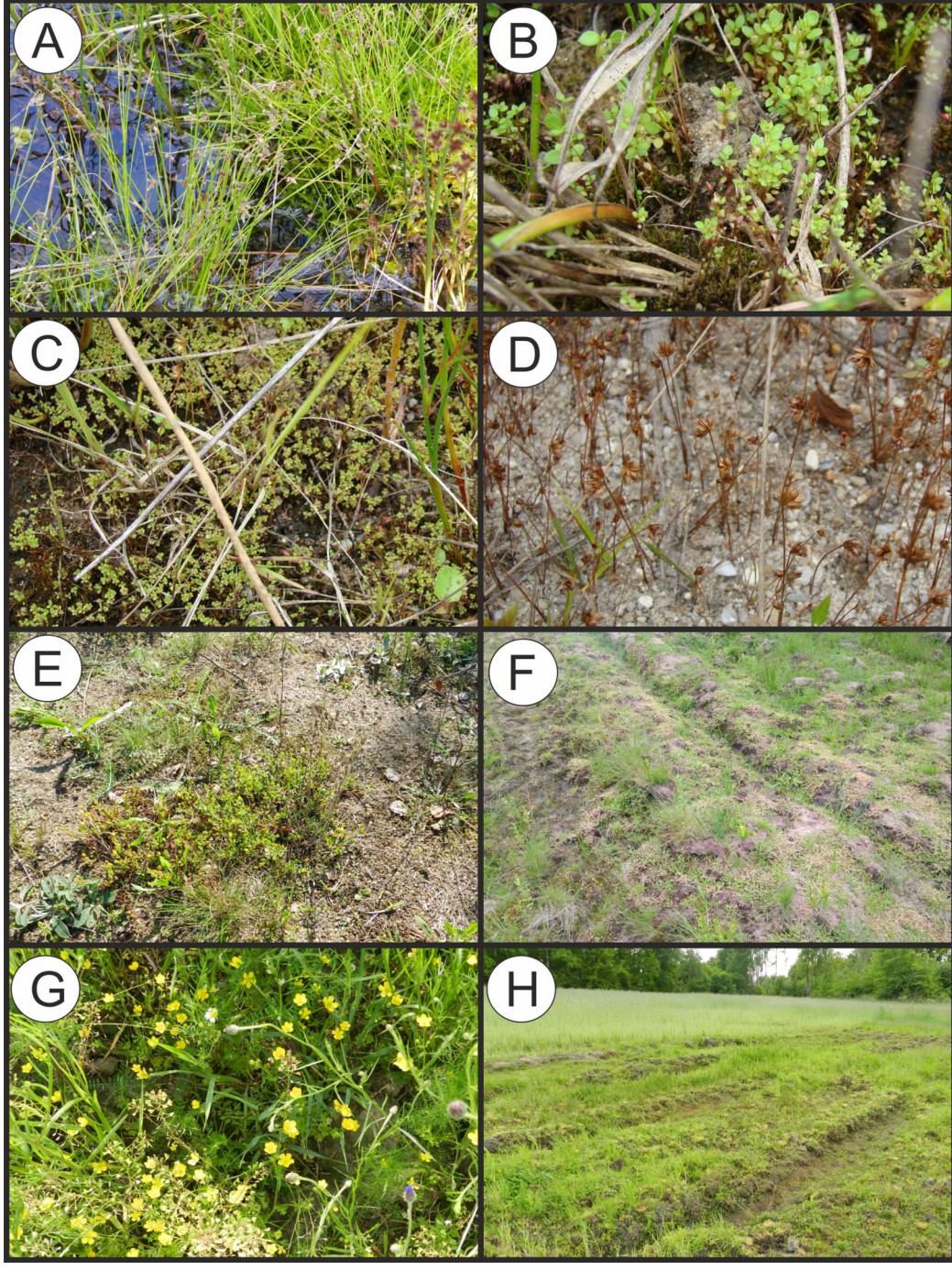

**Figure 11 Vegetation of the Radiolion alliance.** (A) Phytocoenose of the Stellario uliginosae-Isolepidetum setaceae association. (B, C, D) Phytoceonoses of the Centunculo minimi-Anthoceretum punctati association. (E) A patch of the Hyperico humifusi-Spergularietum rubrae. (F) Phytocoenose of the Panico-Illecebretum verticillati association. (G) A patch of Cerastio dubii-Ranunculetum sardoi. (H) A community of *Montia arvensis* (Photos by S. Rosadziński).

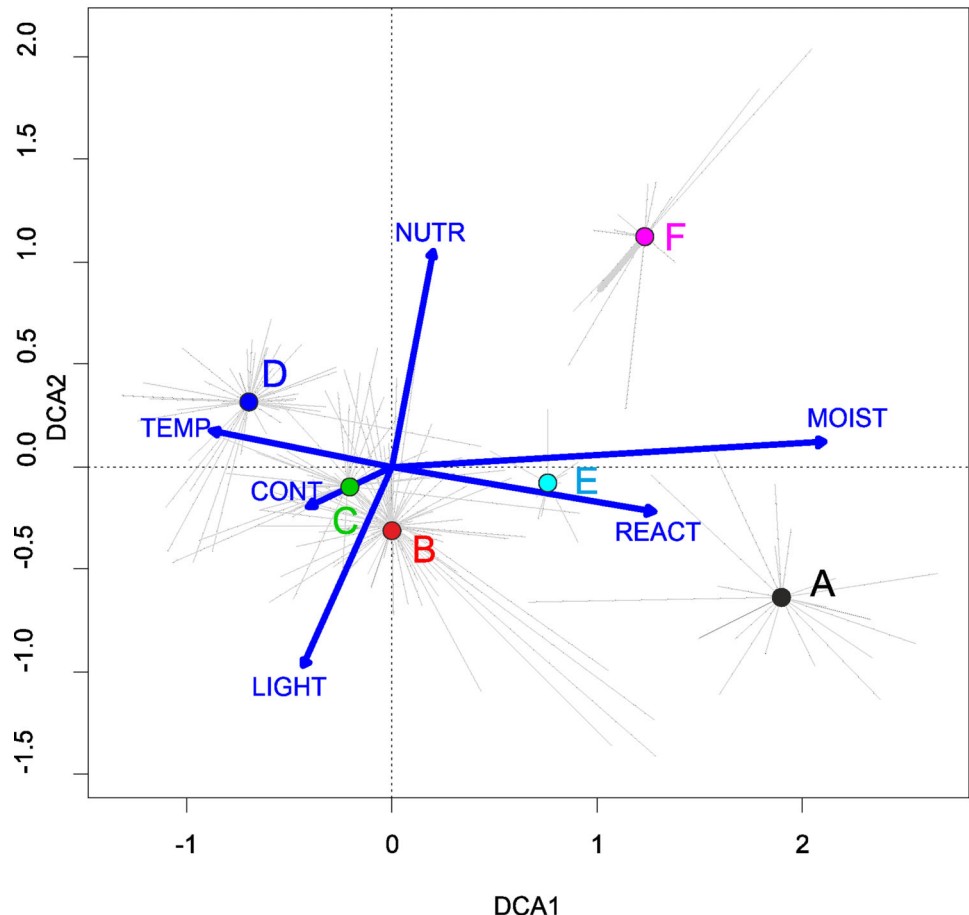

**Figure 12 DCA ordination diagram of samples of Radiolion linoidis plant communities.** The first two ordination axes explain 26.8% and 36.3% of the total species-environment relations variability, and additionally explain 5.5% and 9.0% of the total species variability ($n$ = 223). Colored dots mark the centroids. (A) Stellario uliginosae-Isolepidetum setaceae. (B) Centunculo minimi-Anthoceretum punctati. (C) Hyperico humifusi-Spergularietum rubrae. (D) Panico-Illecebretum verticillati. (E) Cerastio-Ranunculetum sardoi. (F) Community with *Montia arvensis*.

*Plantago major subsp. intermedia, Juncus capitatus, Rumex acetosella, Gypsophila muralis, Equisetum arvense, Spergula arvensis, Hypericum humifusum*
**Dominant species**: *Juncus bufonius, Anthoceros punctatus s. l. l.*

**Distribution:** The Centunculo minimi-Anthoceretum punctati occurs mostly in the south-eastern part of the country: the Masovian Plain, the Małopolska Upland, the Podlasie Lowland and the Beskid Foothills. Additionally, a single site was noted in the western Poland—Fig. 10.

**Physiognomical layout:** The communities are characterized by small annual vascular plants and significant share of bryophytes (Fig. 11).

**Habitat requirements:** The community develops on arable fields, in the second half of summer and early autumn. A necessary condition for its development is the presence of an exposed and moist substrate. In such places, rainwater stays longer, and there are no dense

high weeds stands. These are furrows and mid-field depressions, drying edges of puddle margins, wet spots on clay or sandy, slightly acidic soil. This association has lower moisture and higher temperature demand within the Radiolion alliance (Fig. 12).

### Hyperico humifusi-Spergularietum rubrae *Wójcik, 1968*

**Diagnostic species**: *Spergularia rubra, Gypsophila muralis, Hypericum humifusum, Anagallis minima, Anthoceros punctatus s. l., Laphangium luteoalbum,*
**Differential species:** *Scleranthus annuus, Convolvulus arvensis, Veronica serpyllifolia, Anthemis arvensis, Rumex acetosella, Elytrigia repens, Sagina procumbens, Oxalis fontana, Anagallis arvensis, Setaria pumila, Polygonum aviculare s. l., Myosotis arvensis, Raphanus raphanistrum, Erigeron canadensis, Viola arvensis, Crepis tectorum, Spergula arvensis*
**Constant species**: *Spergularia rubra, Gypsophila muralis, Juncus bufonius, Gnaphalium uliginosum, Sagina procumbens, Polygonum aviculare s. l., Scleranthus annuus, Plantago major subsp. intermedia, Rumex acetosella, Hypericum humifusum, Anagallis minima, Elytrigia repens, Veronica serpyllifolia, Setaria pumila, Erigeron canadensis, Spergula arvensis, Anthemis arvensis, Equisetum arvense, Viola arvensis*
**Dominant species**: *Juncus bufonius, Rumex acetosella*

**Distribution:** The Hyperico humifusi-Spergularietum rubrae occurs in the south-eastern part of Poland (Fig. 10).
**Physiognomical layout:** The association dominated by small annual vascular plants, mostly diagnostic for Radiolion alliance with considerable constancy and also has species diagnostic for Papaveretea vegetation (Fig. 11).
**Habitat requirements:** The Hyperico humifusi-Spergularietum rubrae occurs in conditions very similar to Centunculo minimi-Anthocetetum punctati and it is found in wet depression in arable fields and in vicinity of mid-field water reservoirs. Likewise, DCA analysis results showed similarity in habitat demand as Centunculo minimi-Anthoceretum punctati (Fig. 12).

### Panico-Illecebretum verticillati Diemont et al., 1940

**Diagnostic species**: *Illecebrum verticillatum, Spergularia rubra, Hypericum humifusum, Radiola linoides*
**Differential species:** *Teesdalia nudicaulis, Arnoseris minima, Spergula arvensis, Scleranthus annuus, Holcus mollis, Rumex acetosella, Digitaria ischaemum, Raphanus raphanistrum, Setaria pumila, Fallopia convolvulus, Aphanes microcarpa, Anthoxanthum aristatum, Galeopsis tetrahit s. l., Rhinanthus angustifolius, Bidens tripartitus, Setaria viridis, Stachys palustris, Persicaria hydropiper, Mentha arvensis, Viola tricolor agg., Myosotis arvensis, Achillea millefolium*
**Constant species**: *Illecebrum verticillatum, Spergula arvensis, Rumex acetosella, Juncus bufonius, Spergularia rubra, Scleranthus annuus, Persicaria hydropiper, Hypericum humifusum, Gnaphalium uliginosum, Radiola linoides, Mentha arvensis, Setaria pumila, Teesdalia nudicaulis, Bidens tripartitus, Arnoseris minima, Polygonum aviculare s. l., Equisetum arvense, Fallopia convolvulus, Raphanus raphanistrum, Agrostis stolonifera*

**Dominant species**: *Illecebrum verticillatum*

**Distribution:** Panico-Illecebretum verticillati occurs in the south-eastern part of Poland (Fig. 10) similarly to Centunculo minimi-Anthocetetum punctati and Hyperico humifusi-Spergularietum rubrae.

**Physiognomical layout:** It develops at the end of summer, in the form of small patches, on moist soil in crop stands or on stubble. Compared to Centunculo-Anthoceretum, it differs in its abundant occurrence of *Illecebrum verticillatum* and *Spergula arvensis* and others sandy, acidophilous plants with the negligible amount of bryophytes.

**Habitat requirements:** The soil is more acidic; according to DCA, it also has lower humidity and higher temperature requirements (Fig. 12).

## Cerastio dubii-Ranunculetum sardoi Oberdorfer ex Vicherek 1968

**Diagnostic species:** *Ranunculus sardous, Myosurus minimus, Plantago major subsp. intermedia*

**Differential species:** *Capsella bursa-pastoris, Bryum dichotomum, Bryum caespiticium, Bryum ruderale, Papaver rhoeas, Linaria spartea, Galium aparine, Dicranella varia, Thlaspi arvense, Veronica persica, Barbula unguiculata, Veronica hederifolia, Erodium cicutarium, Valerianella locusta, Tortula modica, Phascum cuspidatum, Brachythecium rutabulum, Geranium pusillum, Ceratodon purpureus, Stellaria media s. l., Marchantia polymorpha, Leptobryum pyriforme, Vicia hirsuta, Tripleurospermum inodorum agg.*

**Constant species**: *Ranunculus sardous, Plantago major subsp. intermedia, Myosurus minimus, Juncus bufonius, Gnaphalium uliginosum, Capsella bursa-pastoris, Stellaria media s. l., Ceratodon purpureus, Bryum dichotomum, Bryum caespiticium, Tripleurospermum inodorum agg., Dicranella varia, Bryum ruderale, Viola arvensis, Vicia hirsuta, Veronica persica, Thlaspi arvense, Sagina procumbens, Riccia glauca, Polygonum aviculare s. l., Mentha arvensis, Leptobryum pyriforme, Galium aparine, Erodium cicutarium, Bryum argenteum, Brachythecium rutabulum*

**Dominant species**: *Ranunculus sardous, Myosurus minimus, Riccia glauca, Lythrum portula, Juncus bufonius, Gnaphalium uliginosum*

**Distribution:** The association found on scattered localities mainly in river valleys across Poland (Fig. 10).

**Physiognomical layout:** The patches of the association develop in early summer—mainly in June and July. The number of species from the Isoëto-Nanojuncetea class is insignificant, and plants of the Papaveretea class are considerable contributors (Fig. 11).

**Habitat requirements:** It occurs in damp places in mid-field furrows, on fallow fields and their margins. It is a short-term seasonal community, composed exclusively of annuals. Analysis of Ellenberg indicator values showed that Cerastio dubii-Ranunculetum sardoi has a higher demand for moisture and nutrient content (Fig. 12).

## Community with *Montia arvensis*

**Diagnostic species**: *Montia arvensis, Myosurus minimus, Riccia sorocarpa*

**Differential species**: *Dicranella staphylina, Holcus lanatus, Epilobium ciliatum, Lotus pedunculatus, Bryum subapiculatum, Lathyrus pratensis, Bryum violaceum, Lysimachia vulgaris, Myosotis discolor, Pohlia annotina, Poa trivialis, Arabidopsis thaliana, Cerastium glomeratum, Eupatorium cannabinum, Vicia hirsuta, Solidago gigantea, Rumex crispus, Lamium purpureum, Bidens frondosus, Veronica arvensis, Ranunculus repens, Stellaria palustris, Pleuridium acuminatum, Glyceria declinata, Galium palustre agg., Tripleurospermum inodorum agg., Leptobryum pyriforme, Ceratodon purpureus*

**Constant species**: *Montia arvensis, Juncus bufonius, Myosurus minimus, Lotus pedunculatus, Holcus lanatus, Vicia hirsuta, Tripleurospermum inodorum agg., Ranunculus repens, Gnaphalium uliginosum, Epilobium ciliatum, Dicranella staphylina, Bidens frondosus, Agrostis stolonifera, Veronica arvensis, Rorippa palustris, Persicaria hydropiper, Lysimachia vulgaris, Polygonum aviculare s. l., Poa trivialis, Plantago major subsp. intermedia, Persicaria lapathifolia s. l., Mentha arvensis, Ceratodon purpureus, Bryum subapiculatum*

**Dominant species**: *Montia arvensis, Juncus bufonius, Pohlia annotina, Argentina anserina*

**Distribution:** This community was recorded in western Poland in the Lubuskie Region only (Fig. 10).

**Physiognomical layout:** *Montia arvensis* is mostly accompanied by *Myosurus minimus* and common segetal weeds of the Papaveretea class (Fig. 11).

**Habitat requirements:** Community with *Montia arvensis* was found in moist, sandy arable fields. It develops mostly near backwaters after spring thaws, on silts, in drainage ditches and spring outflows. Less frequently it was found in fallows and wastelands. According to DCA, the association showed relatively high soil reaction and moisture (Fig. 12).

## DISCUSSION

The distribution of the stands of Isoëto-Nanojuncetea class in Poland reflects the following patterns: they are quite common in the central and southern part of Poland, rarely found in the east and central Pomerania and absent in Warmia and Mazury regions. This is because 25 (65.7%) characteristic taxa of the class extended to the eastern distribution boundary of their European range (*Popiela, 2005*). Moreover, a large part of these communities in Poland has no characteristic feature and could be classified into alliances only.

In accordance with the syntaxonomic concept of *Mucina et al. (2016)*, the order Nanocyperetalia Klika 1935 includes six alliances, of which three occur in Central Europe: Eleocharition soloniensis Philippi 1968, Verbenion supinae Slavnić 1951 and Radiolion linoidis *Pietsch, 1973*. All occur in Poland (*Kącki, Czarniecka & Swacha, 2013*). Two alliances are the most widespread - the Eleocharition soloniensis (53.3% of relevés fall into this alliance) and the Radiolion linoidis (39.3% relevés). The Verbenion supinae vegetation represents only 7.3% of the relevés set. Our results showed that vegetation of Radiolion is likely more frequent in Poland than in the southern part of Europe, especially in the Czech Republic and Slovakia (*Šumberová & Hrivnák, 2013*). The Radiolion linoidis is mostly distributed in the Atlantic zone of Europe and additionally in damp

habitats of heaths (*Moor, 1936*, *Pietsch, 1973*). Habitats of this vegetation are to highest extend, exclusively moistened by precipitation occasionally by floodings, as reported by *Pietsch (1973)*. According to *Deil (2005)* and *Šumberová & Hrivnák (2013)*, high amount of precipitation before growing season is crucial for the development its characteristic species and the vegetation itself. Our results showed that the Radiolion communities require less humidity compared to the other two alliances. In the eastern and central parts of Poland the vegetation of Radiolion alliance is more common (traditional, family farming system prevails there). It rarely occurs in the northern and western part of the country, where formerly socialized agriculture prevailed (*Sikora, 2012*). In most of vegetation's overviews of Central European the Verbenion supinae vegetation was not reported (*Oberdorfer, 1957*, *1977*; *Täuber & Petersen, 2000*; *Matuszkiewicz, 2007*); however, it has recently been indicated in Czech Republik and Slovakia (*Šumberová & Hrivnák, 2013*).

A limited number of diagnostic species of Verbenion supinae alliance found in Poland. Its occurrence was uncertain so far (*Popiela, 1997*). It is not clear the relation between Verbenion supinae and Nanocyperion in Europe. The vegetation of Verbenion was considered as Mediterranean and subcontinental, sub-halo-nitrophilic communities and Nanocyperion as Euro-Siberian vegetation on acidophilous to neutrophilous soils (*Deil, 2005*; *Biondi et al., 2014*). On the other hand Verbenion is considered as the vegetation of nemoral zone of Central and southeastern Europe while Nanocyperion of the submediterranean and Atlantic regions of Europe (*Mucina et al., 2016*). We followed *Mucina et al. (2016)* concept of the class Isoëto-Nanojuncetea, although the Verbenion supinae alliance in Poland is strongly impoverished.

Altogether, 14 assemblages were distinguished using the formal definition in Poland. The most widespread are associations Polygono-Eleocharitetum ovatae and Cypero-Limoselletum. This also corresponds with the observations from other European countries (e.g. *Phillipi, 1977*; *Müller-Stoll & Pietsch, 1985*; *Bagi, 1988*; *Täuber, 2000*; *Šumberová & Hrivnák, 2013*). We described Cypero-Limoselletum which was included as synonym into association Cyperetum micheliani (*Šumberová, 2011a*). In Poland *Cyperus michelianus* is very rare sub-mediterranean species occurring on the north limits of its geographical range. We distinguished this association based on relevés from the single locality in Poland. Following *Horvatić's (1931)* concept plots with dominance of *Cyperus michelianus* were included only. Most common in Poland is the Cypero-Limoselletum association described after Oberdorfer by *Korneck (1960)*. It is well defined pionier vegetation mostly on natural habitats along big rivers banks in western part of the country. Cypero-Limoselletum is widespread in Europe and Asia (e.g. *Pietsch, 1973*; *Rašomavičius & Biveinis, 1996*; *de Foucault, 2013*; *Dubyna et al., 2015*; *Taran, 2019*), it was also classified as a community *Cyperus fuscus-Limosella aquatica* (*Popiela, 1997*). Cypero-Limoselletum is well separated vegetation units of cyperoids, rarely enriched (especially of exposed pond bottoms) by diagnostic species of Polygono-Eleocharitetum. Among the Eleocharition alliance we also described a community with *Coleanthus subtilis* which recently found in Poland (*Fabiszewski & Cebrat, 2003*). It occurs preferably in ponds used in the annual or biennial cycle (stocking ponds), with emptying in late autumn and filling at the turn of April and May. The habitat conditions of plant communities with

*Coleanthus subtilis* showed that it develops on organic soils with a pH of 6.4–7.8, rich in N, K, Mg, Ca and Na, and low in P (*Dajdok et al., 2017*). In Europe, there is no uniform approach to the classification of patches with *Coleanthus subtilis*. They were included in Polygono-Eleocharitetum, Cyperetum micheliani and Stellario uliginosae-Isolepidetum setaceae (*Pietsch, 1973*; *Brullo & Minissale, 1998*; *Šumberová & Hrivnák, 2013*; *Richert et al., 2016*).

Communities of Verbenion supinae alliance require higher contents of calcium and soluble mineral salts in the substrate (*Pietsch, 1973*). They were rarely reported from Poland. The Pulicario vulgaris-Menthetum pulegii was documented mainly by *Strech (1941)* and *Borysiak (1994)* and included into Bidentetea class. According to our results, relevés with high number of annuals of Isoëto-Nanojuncetea class are classified to this unit. Veronico anagalloidis-Lythretum hyssopifoliae still poorly recognized in Poland, probably has a transitional character to halophitic communities of the Festuco-Puccinellietea class and requires further research.

Cyperetum flavescentis is classified differently in the phytosociological literature (*Moor, 1936*; *Pietsch, 1973*; *Pietsch & Müller-Stoll, 1974*). Due to data scarcity, *Popiela (1997)* temporarily included this association into the alliance Eleocharition soloniensis. Cyperetum flavescentis was rarely found in Poland. However, new findings of large populations of *Cyperus flavescens* have recently been reported from the south and north-eastern parts of Poland (*Marciniuk et al., 2020*). Habitats of this species are periodically wet, and soil are alkaline or slightly acidic. More frequent warm summers with a heavy rainfall in Poland enhance the population of this species (*Marciniuk et al., 2020*). The authors suggested that geographical range of *Cyperus flavescens* in Europe will increase because of global warming. In Central Europe Cyperetum flavescentis was rarely recorded in until now (*Pietsch, 1973*; *Pietsch & Müller-Stoll, 1974*; *Popiela, 1997*). It is therefore poorly represented in our dataset but it is clearly distinguishable by the dominance of *Cyperus flavescens* and accompanying species, mostly marsh and meadows species. Currently increasing population in Poland is crucial for its preservation in Central Europe (*Marciniuk et al., 2020*). It is important because *C. flavescens* is listed on the IUCN red list as a species of lowest risk of extinction (*Lansdown, 2018*) and is endangered in many European countries (*Hodálová, Feráková & Procházka, 1999*; *Korneck, Schnittler & Vollmer, 1996*; *Nikolić & Topić, 2005*; *Kaźmierczakowa et al., 2016*; *Grulich, 2017*).

The Eleocharito-Schoenoplectetum supini association with diagnostic species such as *Elatine alsinastrum*, *Schoenoplectus supinus* and *Lythrum hyssopifolia* has a distinct floristic and ecological layout. Our results correspond with the outcomes revealed by *Šumberová & Hrivnák (2013)*, which included plots with *Elatine alsinastrum* in the Verbenion supinae. We use the name Eleocharito-Schoenoplectetum supini because of the priority of the valid name published by *Ubrizsy (1948)*. This association was found on temporary ponds in mid-field depressions that occurred frequently on the uplands of Lublin region in Poland (*Krawczyk et al., 2016*). Formerly plots with *Elatine alsinastrum* accompanyied with *Juncus tenageia* were classified to Elatino alsinastri-Juncetum tenageiae, and placed in the Eleocharition soloniensis alliance (e.g. *Pietsch, 1973*; *Täuber, Bruns & Steinhoff, 2007*) or left rankless (*Kępczyński & Rutkowski, 1991*; *Popiela & Fudali, 1996*;
*Krawczyk et al., 2016*). Moreover, the Elatino alsinastri-Juncetum tenageiae was considered as a synonym of Junco tenageiae-Radioletum linoidis association which was included in Radiolion alliance (*Šumberová, 2011b*). In addition, in Pannonian Basin communities with *Elatine alsinastrum* were classified to Eleocharito-Schoenoplectetum supini or Elatinetum alsinastri (*Nagy et al., 2009*; *Hrivnák & Slezák, 2017*). In our data a high constancy of *Alisma lanceolatum* characterize the Eleocharito-Schoenoplectetum supini. This species is reported as closely related to Eleocharito palustris-Alismatetum lanceolati, which is a littoral, perennial community (*Hrivnák et al., 2015*). According to the formal definition of the Eleocharito-Schoenoplectetum supini, relevés with high amount of *Alisma lanceolatum* and without species from Isoëto-Nanojuncetea were excluded and considered as rush vegetation. In general, communities of this type are poorly understood in Europe, due the fact that they occur infrequently and often in man made habitats. Studies have shown that they can remain dormant for a very long time, even for several decades (*Täuber, Bruns & Steinhoff, 2007*; *Lukács, Sramkó & Molnár, 2013*). During favorable conditions, their fast development is possible due to a very durable and abundant soil seed bank (*Albrecht et al., 2019*). The communities of this group are more associated with continental climate, for example in Hungary, where they are more common and diverse (*Lukács, Sramkó & Molnár, 2013*). In Poland, they occur sporadically on the north-western range limit.

Among the Radiolion alliance the Centunculo-Anthoceretum punctati, the Panico-Illecebretum verticillati and the Hypericum humifusi-Spergularietum rubrae were found to be the most widespread associations. They are all distributed in the south-eastern part of Poland and develop mostly in wet furrows of arable fields. The Hyperico humifusi-Spergularietum rubrae was described by *Wójcik (1968)*, but *Popiela (1997)* pointed out that this association was similar to the Panico-Illecebretum verticillati. In the Czech Republic Hyperico humifusi-Spergularietum rubrae was included in Centunculo-Anthoceretum punctati (*Šumberová & Hrivnák, 2013*). According to our results, both associations are clearly distinct in Poland, despite they share many diagnostics species. In Radiolion alliance, we found a new community with *Montia arvensis* developed on arable fields in the west part of the country. Vegetation with the dominance of *Montia arvensis* and *Myosurus minimus* has been classified so far to Centunculo-Anthoceretum punctatae, Cicendietum filiformis or Molineriello-Illecebretum verticillatae (*Pietsch, 1973*; *Brullo & Minissale, 1998*). This community seems to be well distinguished from other Radiolion linoidis communities, however, further research is required, among others a description of environmental conditions, distribution pattern and, eventually, define the association.

The first formal classification of the Isoëto-Nanojuncetea class in Poland revealed a high diversity of ephemeral vegetation found on their north-eastern distribution limits in Europe. We described two new plant communities within Eleocharition and Radiolion alliance. Although, we based on large data set, some very scarce vegetation units were recognized based on relatively small number of relevés. Hence, further research focused on rare and conservationally important ephemeral wetland vegetation is needed to present the final classification of this type of vegetation in Poland and to enable its effective conservation. Nevertheless, the results of our research show for the first time the

comprehensive, reproducible in the context of analyzes, with the use of the largest database, classification that is consistent with neighboring areas.

## ACKNOWLEDGEMENTS

We are thankful to Mr. Marcin Dec (Florida, USA) for the linguistic revision of the manuscript. We are very grateful to Professor Ulrich Deil and to an anonymous reviewer for all comments and suggestions included in the manuscript.

### Funding

The research has been financially supported by the Polish Ministry of Science and Higher Education through grants no. NN3 04292840 (Stanisław Rosadziński) and 3 PO4C0352 (Agnieszka Anna Popiela). There was no additional external funding received for this study. The funders had no role in study design, data collection and analysis, decision to publish, or preparation of the manuscript.

### Grant Disclosures

The following grant information was disclosed by the authors:
Polish Ministry of Science and Higher Education: NN3 04292840 and 3 PO4C0352.

### Competing Interests

The authors declare that they have no competing interests.

### Author Contributions

- Zygmunt Kącki conceived and designed the experiments, performed the experiments, analyzed the data, prepared figures and/or tables, authored or reviewed drafts of the paper, and approved the final draft.
- Andrzej Łysko conceived and designed the experiments, performed the experiments, analyzed the data, prepared figures and/or tables, authored or reviewed drafts of the paper, and approved the final draft.
- Zygmunt Dajdok conceived and designed the experiments, performed the experiments, authored or reviewed drafts of the paper, and approved the final draft.
- Piotr Kobierski conceived and designed the experiments, performed the experiments, authored or reviewed drafts of the paper, and approved the final draft.
- Rafał Krawczyk conceived and designed the experiments, performed the experiments, authored or reviewed drafts of the paper, and approved the final draft.
- Arkadiusz Nowak conceived and designed the experiments, performed the experiments, authored or reviewed drafts of the paper, and approved the final draft.
- Stanisław Rosadziński conceived and designed the experiments, performed the experiments, authored or reviewed drafts of the paper, and approved the final draft.
- Agnieszka Anna Popiela conceived and designed the experiments, performed the experiments, analyzed the data, authored or reviewed drafts of the paper, and approved the final draft.

## Data Availability

The raw measurements are available in the Supplemental Files:

(A) ESY_FILE_ISOETO_NANO.txt: file for expert system for automatic classification ephemeral wetland vegetation (Isoëto-Nanojuncetea class) in Poland (Central Europe) - created for JUICE software

(B) Supporting_information_2.xlsx: Synoptic table with diagnostic species for associations. Percentage occurrence and phi coefficient values are given. Species with less than 5 occurrences (except diagnostic and differential species) were removed.

## Supplemental Information

Supplemental information for this article can be found online at http://dx.doi.org/10.7717/peerj.11703#supplemental-information.

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
