# Peer review of "Formalized classification of ephemeral wetland vegetation (Isoëto-Nanojuncetea class) in Poland (Central Europe)"

_PeerJ, doi:10.7717/peerj.11703_

## Round 0.1 · original submission · Minor Revisions

Dear Dr. Katcki, dear Dr.Popiela,

We received two reviews of your manuscript.

Both reviewers agree that this study can be published after several minor corrections.

I also agree with these evaluations, but I recommend that you consider them carefully. The quality of the language must be improved. Please consider that the iteration according to minor revision indication does not make a guarantee of publication.

Sincerely,

Leonardo Montagnani

·

Basic reporting

The English text of the final version should be checked carefully. A number of misspellings are annotated in the attached pdf-version

all other items o.k.

Experimental design

The authors analysed all the relevés recording ephemeral dwarf vegetation in Poland, which can be classified into the Isoeto-Nanojuncetea class. Releves have been selected from the Polish Vegetation data base on the basis of pre-selected 43 diagnostic species. The paper gives an excellent overview over this vegetation type and habitat, sheltering a high number of vulnerable plant species.

Aims of the study are clearly formulated, methods explained in detail. Statistical treatment (a.o. with Cocktail method) is adequate. The accepted syntaxa at the levels of alliance and association result from a logic and objective classification process using sociological species groups. The data are documented in a synoptic constancy table Tab S2, distribution in Poland is given in a series of maps.

Validity of the findings

no comments, but see general comments for the authors

Additional comments

There are two major points, authors should take into consideration for the revision:
1. I would hesitate to describe the stands with Coleanthus subtilis as a new association (lines 615-619) and would prefer to classify it as a rankless community or dominance community. It is of course a clearly separated cluster in your data set, but this new association is only of local relevance in a very small part of Poland. C. s. is scattered locally over Europa, Asia and North America, and can be an ephemeral, locally dominant species associated with different species and occur under a variety of environmental conditions (see Richert et al. 2016). It occurs in various Eleocharition and Nanocyperion communities. When describing it formally as ass. nova, you should compare it with other syntaxa, just to mention in Eleocharito-Caricetum bohemicae in Austria (Koch et al. 2005, Bernhardt et al. 2005), Coleantho-Spergularietum echinetosum Hejny 1978 in CZ, Chenopodio rubri-Coleanthetum subtilis Le Bail et al. 2013 in Foucault 2013 in France, Rorippo-Limoselletum coleanthetosum Taran 2005 in Western Siberia, with Limosella aquatica in Kanada (Catling 2009)
2. You should make clearer in chapters results and discussion, that the differentiating and explaining environmental factors (see ordination diagrams) are concluded from the Ellenberg Indicator Values of the considered species and are not based upon observations or measurements on the plots. And you should discuss, that other ecological factors (not available for the plots) might be more relevant (and less “proxy factors”) as selective filters for the species assemblages (just to mention a few: hydroperiod, duration and depth of submergence, surface temperature during emergence and germination of the annuals etc).

Recommendation for authors and editors: To my opinion, it would be better for the reader to shift the constancy table Tab S2 from the electronic supplement into the printed version

minor comments
Line 127: The number of finally used relevés (816) should be mentioned in the abstract
Lines 180-182 belong to methods
Lines 180-182 refer to Tab 1 in text here
please give the arguments why you propose a new name for Digitario-Illecebretum and refer to the new version of ICPN (Theurillat et al. 2020)

Reviewer 2 ·

Basic reporting

The revised manuscript “Formalized classification of ephemeral wetland vegetation
(Isoëto-Nanojuncetea class) in Poland (Central Europe) ” is generally well-written, clear, and in my opinion definitely worth to be published. Figures and Tables are necessary and well designed.
This study, although it has rather the nature of checklist, presents interesting data on vegetation classification of rare, protected and definitely under examined plant vegetation from Isoëto-Nanojuncetea class. The authors analysed 1340 relevés (of almost 70000!) that were classified to the I-N class, as well as described and discussed 13 associations (among them Coleanthetum subtilis as a new syntaxon). Besides, they presented the maps of distributions of the associations in Poland and statistical-ecological analyses of the analysed plant communities. Of all the reasons, I find this manuscript very interesting and useful. It share the light on the syntaxonomy of this unique vegetation in Poland (central Europe) and the results will be adopted and used in conservation strategies both in Poland and European Union, having in mind that the habitats is protected under the Natura 2000 programme.

Experimental design

Research questions were well defined and the ms fills the knowledge gap in current and proper taxonomy of this plant vegetation.

Validity of the findings

no comments

Additional comments

The manuscript, in some parts needs supplementation

In my opinion, cluster analyses of the analysed syntaxon should be presented, to show the readers general ordination of this ephemeral wetland vegetation in Poland (in addition to analyses of particular orders).

Particular remarks
Abstract:
Reliable and formalized classification of the class Isoëto-Nanojuncetea has not been
performed in Poland. – I think that it is NOT true! (see works of A. Popiela in References, as well as Kornaś (1960), Wójcik (1968) and Zając (1988)) Please remove or rework this phrase.

line 56 - defined. examples are vegetation of the class Isoëto-Nanojuncetea – remove dot.
line 66 – I think that the most recent literature date on classification of Isoëto-Nanojuncetea should be added.
line 76 - Plant communities of Isoëto-Nanojuncetea class in Polish territory reaches – replace with Plant communities of Isoëto-Nanojuncetea class in Poland reaches
line 86 – add Bruelheide, H., et al 2019. sPlot – a new tool for global vegetation analyses. – Journal of Vegetation Science. 30(2): 161-186. https://doi.org/10.1111/jvs.12710
In the Introduction the authors should mentioned on conservation and the importance of this vegetation in European Union. The threats of particular communities should be also presented/discussed in the MS either in description of particular syntaxa or in the Discussion
Results should starts from the general ordination analysis of the studied vegetation type/analysed vegetation plots. In the description of particular syntaxa, a short information on previous studies of these vegetation will be welcome. Now it looks like all of these associations are studied here for the first time.
line 194 In Poland, the Eleocharition ovatae alliance includes three plant communities – replace included with comprises
line 187 Differential – it will be easier to readers if the taxa in Differential, Constant and Dominant subsubchapters will be presented in alphabetic order
line 188 Constant species: Cyperus fuscus, - The nomenclature of species in the MS should be checked, here is Cyperus and afterwards Pycreus – so please be consequent. Moreover you already stated that you used Euro-Med-Plant+ nomenclature for species names.
line 228 Cyperetum micheliani Horvatić 1931 – I am wonder if this name is OK? I think that the authors should explain why they have no Cyperus micheliana in diagnostic species in Notes. Is this name Cyperetum micheliani shouldn’t be replaced by more actual?
line 253-254 - Coleanthetum subtilis ass. nova hoc loco; Lectotypus hoc loco designatus: Dajdok et al. (2017) – I do not understand it. This plant ass has been described by the authors in this MS ass. nova hoc loco! – so why you decided to cite lectotype instead of holotype? Or maybe you are going to describe it after Dajdok??? as Coleanthetum subtilis Dajdok ex Kącki et al. ??? I am not familiar with such nomenclature, but it should be explained in the Notes (after Habitat requirements). In Phytotaxa there is a paper on current position of Coleanthus (Nosov et al.) – maybe it also will be useful.
317 Cyperetum flavescentis Koch 1926 - Diagnostic species: Pycreus flavescens - - I think that in this case the name of association could be replaced with Pycreetum flavescentis??
363 Eleocharito-Schoenoplectetum supini Soo & Ubrizsy 1948 nomina inversa prop. – please explain in Notes how it was earlier, and why this nomina has been proposed. By the way souldn’t be now ‘Eleocharito-Schoenoplectelietum supini??’
429 Distribution: Stellario-Isolepidetum – please do not use abbreviations here and leater.
441 Centunculo minimi-Anthoceretum punctati Koch ex Libbert 1932 – I am wonder if it shouldn’t be now Anthocero punctati-Anagaletum minimi Koch ex Libbert 1932 or Anagalo minimi-Anthoceretum punctati Koch ex Libbert 1932??
495 Digitario ischaemi-Illecebretum verticillati Diemont & al. 1940 nom. mut. propos. – please explain in Notes why
582 European range (Popiela 2005). Moreover, a large part of - - Moreover, a majority of

---

## Round 0.2 · Minor Revisions

We received two evaluations of your revised manuscript. A few changes are still asked and detailed by the reviewers. Please consider them carefully in the revised version of your manuscript, which I expect quite soon.

Sincerely,

Leonardo Montagnani

·

Basic reporting

see general recommendations

Experimental design

see general recommendations

Validity of the findings

see general recommendations

Additional comments

the revised version can be accepted after some minor linguistic corrections (see annotations in attached pdf)

Reviewer 2 ·

Basic reporting

no comment

Experimental design

no comment

Validity of the findings

no comment

Additional comments

I find the ms sufficiently improved and it can be published in its current form. I have no further comments to the manuscript. The authors should however consider if the community of Coleanthus subtilis needs to be lectotypified (?).

---

## Round 0.3 · accepted · Accept

Dear Dr Popiela,

I am glad to inform you that I consider your paper acceptable now. Thank you for considering Peerj as the target journal for your studies.

Sincerely,

Leonardo Montagnani